# Systematic analysis reveals the prevalence and principles of bypassable gene essentiality

Jun Li[1], Hai-Tao Wang[1,2], Wei-Tao Wang[1,3], Xiao-Ran Zhang[1], Fang Suo[1], Jing-Yi Ren[1], Ying Bi[1], Ying-Xi Xue[1], Wen Hu[1], Meng-Qiu Dong [1] & Li-Lin Du [1,4]

Gene essentiality is a variable phenotypic trait, but to what extent and how essential genes can become dispensable for viability remain unclear. Here, we investigate 'bypass of essentiality (BOE)' — an underexplored type of digenic genetic interaction that renders essential genes dispensable. Through analyzing essential genes on one of the six chromosome arms of the fission yeast *Schizosaccharomyces pombe*, we find that, remarkably, as many as 27% of them can be converted to non-essential genes by BOE interactions. Using this dataset we identify three principles of essentiality bypass: bypassable essential genes tend to have lower importance, tend to exhibit differential essentiality between species, and tend to act with other bypassable genes. In addition, we delineate mechanisms underlying bypassable essentiality, including the previously unappreciated mechanism of dormant redundancy between paralogs. The new insights gained on bypassable essentiality deepen our understanding of genotype-phenotype relationships and will facilitate drug development related to essential genes.

[1] National Institute of Biological Sciences, 102206 Beijing, China. [2] Chinese Academy of Medical Sciences and Peking Union Medical College, 100730 Beijing, China. [3] College of Biological Sciences, China Agricultural University, 100193 Beijing, China. [4] Tsinghua Institute of Multidisciplinary Biomedical Research, Tsinghua University, 100084 Beijing, China. These authors contributed equally: Jun Li, Hai-Tao Wang, Wei-Tao Wang.  Correspondence and requests for materials should be addressed to L.-L.D. (email: dulilin@nibs.ac.cn)

According to whether gene deletion causes inviability or not, genes can be classified as either essential genes or non-essential genes. Essential genes are considered the foundation for life and gene essentiality is regarded as a key criterion when selecting drug targets for combating pathogens and cancer cells[1,2]. However, essentiality is not a static gene property. In recent years, it has been shown in yeast[3] and human cell lines[4–6] that gene essentiality can vary between genetic backgrounds. Thus, to fully grasp the underpinnings of life and to improve drug target selection, it is imperative to understand which genes can undergo essentiality change and how essentiality change can happen. In particular, it is of both fundamental and practical values to investigate which and how essential genes in a well-defined genetic background can lose essentiality and be converted to non-essential genes.

Essentiality loss can happen through high-frequency spontaneous chromosome copy number changes[7]. Taking advantage of this phenomenon, it has been shown in a systematic study that in *Saccharomyces cerevisiae*, 9% of the essential genes can be rendered dispensable by spontaneous genetic events occurring at ≥ 1% frequencies, which are most probably events altering chromosome copy numbers[8]. These essential genes are termed "evolvable essential genes". For most of the evolvable essential genes, the underlying genetic basis of the essentiality loss, namely, how many and which genes on the chromosome with an altered copy number mediate the essentiality loss, remains unknown. Furthermore, null mutations and missense mutations, which happen spontaneously at frequencies much lower than 1%, cannot be surveyed by this approach. But such mutations are able to suppress the inviability phenotype of essential gene deletions and thus drive essentiality loss. Prominent examples from the literature include suppression of *SEC14* gene deletion via "bypass Sec14" mutations in *S. cerevisiae*[9], suppression of the loss of *MEC1* or *RAD53* by deletion of *SML1* in *S. cerevisiae*[10], and suppression of *cdc25* deletion by either a *wee1* deletion or a *cdc2-3w* mutation in the fission yeast *Schizosaccharomyces pombe*[11,12]. To our knowledge, this type of digenic suppression interaction lacks a name. Because "bypass suppressor" is a term widely used in the genetics literature to refer to the extragenic suppressor of a deletion mutant[13], we name this type of digenic interaction "bypass of essentiality" (BOE). For simplicity, we refer to essential genes that can be rendered non-essential by monogenic suppressors as "bypassable essential genes". Compared to the evolvable gene approach, using experimental means to identify bypassable essential genes and their BOE suppressors can provide a more comprehensive coverage of genes with variable essentiality and can more directly reveal the exact genetic causes of essentiality loss.

Bypassable essential genes have not been analyzed in an unbiased and systematic manner—two previous attempts with limited breadth and depth only identified 4 bypassable essential genes in *E. coli*[14] and 5 bypassable essential genes in *S. cerevisiae*[15]—and thus the true extent of bypassable essentiality remains unknown. Here, we perform a large-scale and unbiased BOE analysis in *S. pombe*, and find that bypassable essential genes are much more prevalent than previously realized. The results of this systematic BOE analysis enable us to identify the general principles as well as functional and evolutionary implications of essentiality bypass. Furthermore, we demonstrate that BOE analysis is especially conducive to inferring functional relationships between genes.

## Results

**Systematic analysis of BOE interactions.** We developed an efficient BOE analysis procedure that uses "query strains" lacking the chromosomal copy of an essential gene ("query gene") but harboring a counter-selectable episomal plasmid containing that gene (Supplementary Figs. 1a-c). To enable the identification of a broad range of suppressor types (Fig. 1a), we induced genetic changes using the chemical mutagen methylnitronitrosoguanidine (MNNG), the transposon *piggyBac* (PB), and an overexpression plasmid library, and termed the BOE suppressors thus obtained C-BOE, T-BOE, and OP-BOE suppressors, respectively (Supplementary Figs. 1b, d). T-BOE and OP-BOE suppressors were exhaustively identified; for query genes having only C-BOE suppressors, we ensured that at least one C-BOE suppressor was identified. We experimentally verified all of the candidate suppressors by independently generating genetic alterations identical or similar to the ones found in the screen hits (Supplementary Fig. 1e).

Aiming to unbiasedly survey the bypassability of essential genes, we targeted the essential genes located on the left arm of chromosome II (chrII-L) and were able to obtain BOE analysis results on 142 (89%) of them (Fig. 1b and Supplementary Data 1); these are a representative set of essential genes (Supplementary Fig. 1 f). The 17 (11%) remaining genes include the telomere protection gene *stn1*, whose efficient bypass by chromosome circularization hindered the search for BOE suppressors[16]. We failed to construct query strains for the other 16 genes, possibly because cells cannot tolerate their altered expression levels when these genes are expressed from plasmids.

In total, we identified and verified 263 BOE interactions that render 38 (27%) of the 142 essential genes dispensable, including all three previously known bypassable essential genes on chrII-L: *res1*, *slx8*, and *rhb1*[17–19] (Fig. 2 and Supplementary Data 2). These

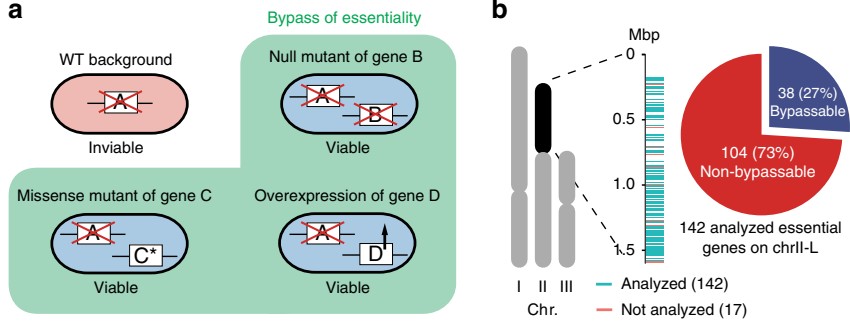

**Fig. 1** Systematic BOE analysis. **a** Three types of monogenic changes that can cause bypass of essentiality (BOE). In the wild-type (WT) background, gene A is essential. BOE may occur when another gene is deleted, mutated, or overexpressed. **b** Systematic BOE analysis for 89% (142/159) of the essential genes on chrII-L in fission yeast led to the identification of BOE suppressors for 27% (38/142) of them; these 38 genes were deemed bypassable. The other 104 genes were deemed non-bypassable

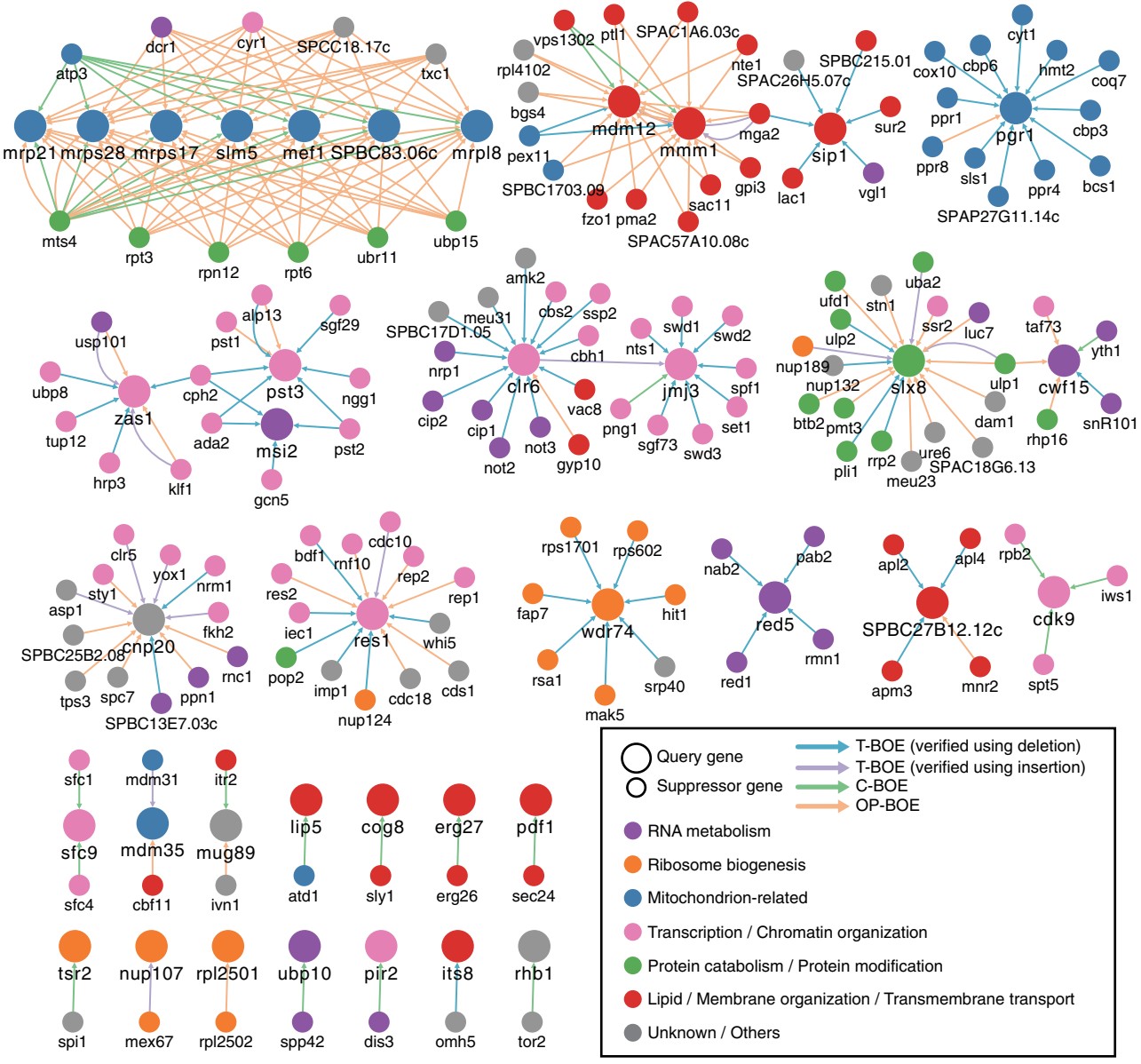

**Fig. 2** BOE interactions that bypass the 38 chrII-L essential genes. Genes are represented as nodes, and BOE interactions are represented as directed edges pointing from suppressor genes (small nodes) to query essential genes (large nodes). Node colors represent the indicated gene functional categories and edge colors represent the indicated suppressor types

BOE interactions, 99% (260/263) of which are previously unknown, connect the 38 query genes (large nodes in Fig. 2) to 157 non-query-suppressor genes (small nodes in Fig. 2), and are classified into four types in Fig. 2: T-BOE suppressors verified using deletion (blue edges in Fig. 2), T-BOE suppressors verified using insertion (purple edges in Fig. 2), C-BOE suppressors (green edges in Fig. 2), and OP-BOE suppressors (orange edges in Fig. 2). Notably, both T-BOE and OP-BOE suppressors were found for 12 query genes, but there is no substantial overlap in suppressor gene identity between these two suppressor types (Supplementary Fig. 1 g), affirming that the use of multiple types of genetic change inducers allows broader coverage of the BOE interaction network.

**Bypassability is inversely correlated with gene importance.** Two gene ontology (GO) slim terms—mitochondrial translation and transcription regulation—are significantly enriched among the bypassable essential genes (Fig. 3a). It makes intuitive sense that essential genes encoding transcription regulators, which by their nature will affect the transcription of only a small subset of genes, have a much higher likelihood to be bypassable than do genes encoding parts of the general transcription machineries (Supplementary Fig. 2a). This led us to hypothesize that bypassable essential genes may be less important than non-bypassable essential genes. This idea may seem unorthodox, because for all essential genes, the growth defect caused by gene deletion—a proxy for gene importance—is the same, namely inviability.

For non-essential genes, less important ones (i.e., those whose deletion causes weaker growth defect) tend to evolve faster[20,21], tend to have lower expression levels, and often employ less optimal codons[22]. We found here that, compared to non-bypassable essential genes, bypassable essential genes have higher evolutionary rates (Fig. 3b), more restricted phylogenetic distributions (Fig. 3c), and less optimal codons (Fig. 3d and Supplementary Fig. 2b). These correlations lend support to the

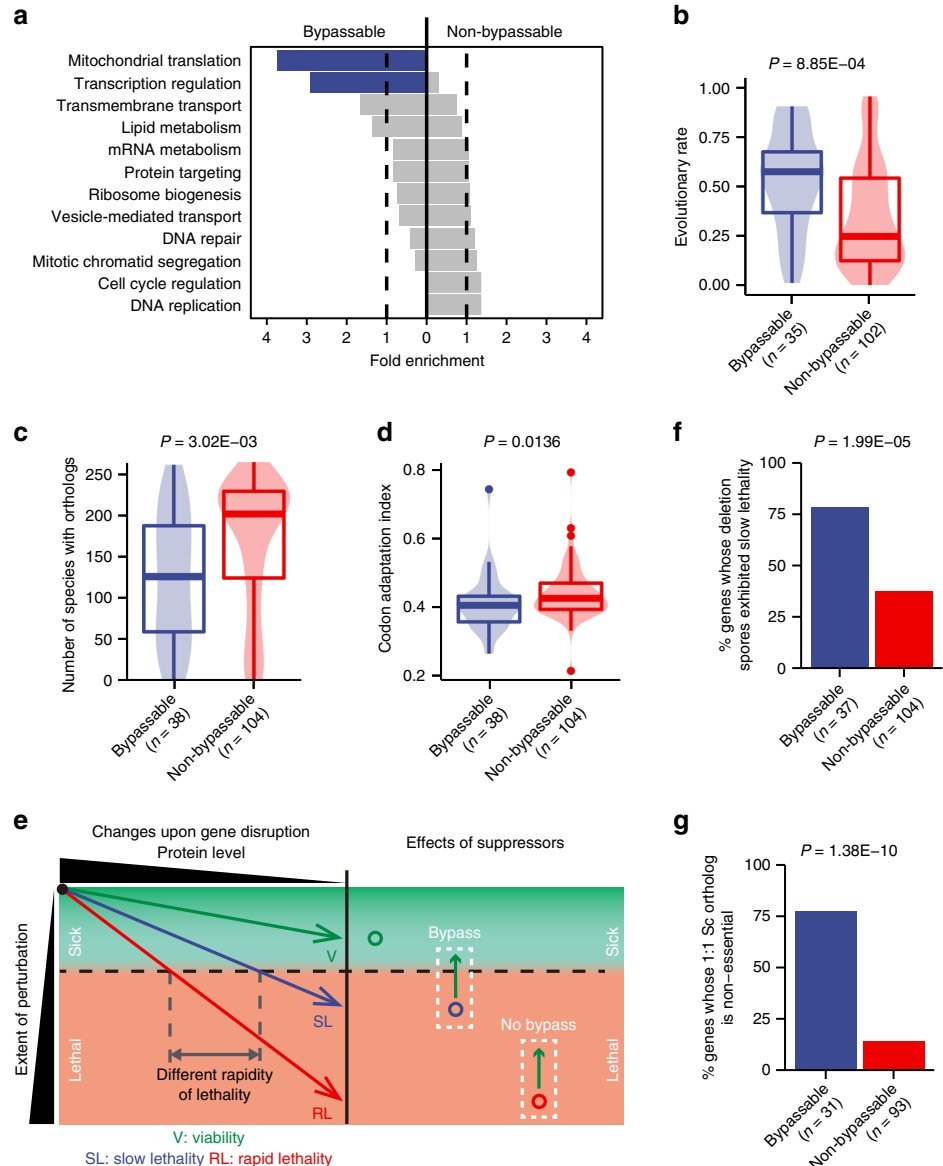

**Fig. 3** Essentiality bypass preferentially occurs to essential genes with lower importance and essential genes exhibiting differential essentiality between species. **a** Two GO slim terms were significantly enriched among bypassable essential genes (Fisher's exact tests, $P = 6.35E-05$ and $0.00146$, respectively, for the two terms at the top). **b–d** Evolutionary rate (**b**), the number of species harboring orthologs (**c**), and codon adaptation index (**d**) exhibit correlations with bypassability (Mann–Whitney–Wilcoxon tests). **e** A model explaining the relationships between gene importance, rapidity of lethality upon gene disruption, and bypassability. Arrowed lines in the left panel show the perturbation becoming more severe as the protein product of the disrupted gene is being depleted. For a non-essential gene (green), the perturbation extent never reaches the lethal threshold (the dashed horizontal line). For essential genes, the threshold is crossed later for a less important gene whose loss results in a weaker perturbation (blue) than for a more important gene whose loss results in a more severe perturbation (red). The former is more likely to be bypassable by ectopic suppressors than the latter (right panel). **f**, **g** Percentages of genes exhibiting slow spore lethality (**f**) and percentages of genes whose one-to-one *S. cerevisiae* ortholog is non-essential (**g**) (Fisher's exact tests). Boxplots show median (centerline), interquartile range (box), and most extreme data points no further than 1.5-fold interquartile range from either end of the box (whiskers)

idea that bypassability is related to gene importance. We therefore propose that the severity of growth-related system perturbation caused by the complete loss of a gene's function—a reflection of gene importance—actually differs between essential genes. Such differences in gene importance may manifest as differences in the rapidity of growth cessation upon gene disruption. To illustrate this point, assuming that growth ceases once the severity of system perturbation reaches a lethal threshold, the threshold would be crossed earlier for genes whose complete loss-of-function results in a more severe perturbation (Fig. 3e, left side).

A natural extension of this model predicts that less important genes—owing to lower degrees of system perturbation caused by gene deletion—are more likely to be bypassable by ectopic suppressors (Fig. 3e, right side).

Indeed, we found that bypassability is strongly correlated with two indications of slow lethality upon gene disruption: the ability of the deletion spores derived from heterozygous deletion diploids to form microcolonies (referred to hereafter as 'slow spore lethality') and high transposon insertion densities when the *Hermes* transposon was used for gene disruption in a pool of

vegetatively growing cells (Fig. 3f and Supplementary Fig. 2c)[23,24]. We ruled out the possibility that slow spore lethality or high *Hermes* insertion densities result mainly from high protein abundance or slow protein turnover rates (Supplementary Figs. 2d-g).

Thus, we conclude that hidden behind the seemingly identical inviability phenotype of essential gene deletions are real differences in gene importance, which manifest as two observable gene properties: rapidity of lethality upon gene disruption and bypassability. Consistent with the idea that gene importance is a key underlying determinant of bypassability, we found that bypassability no longer exhibited statistically significant correlations with evolutionary rate, species distribution, and codon optimality when we controlled for gene importance by considering only genes with slow spore lethality (Supplementary Figs. 2h-j).

**Bypassability is correlated with differential essentiality**. We also examined the relationship between bypassability and the inter-specific variation of gene essentiality by focusing on the 124 query genes that have a one-to-one ortholog in *S. cerevisiae*. Strikingly, among this subset of query genes, 77% (24/31) of the bypassable genes have a non-essential ortholog in *S. cerevisiae*; for the non-bypassable genes this percentage is only 14% (13/93) (Fig. 3g). Thus, bypassability and differential essentiality between these two species are strongly correlated. Examined from a different angle, 37 of these 124 genes have a non-essential ortholog in *S. cerevisiae*, and 65% (24/37) of them can be converted into non-essential genes in *S. pombe* by BOE suppressors. This is remarkable because it means that monogenic changes can eliminate much of the differences in essentiality that have accumulated over the approximately 500 million years since these two species diverged[25]. Interestingly, the correlation between bypassability and differential essentiality remained highly significant after gene importance was controlled for (Supplementary Fig. 2k). In other words, there appears to be a particularly intimate relationship between bypassable essentiality and evolutionary variation of essentiality. It follows that essentiality bypass may be a common cause of essentiality changes during evolution.

**Bypass of the essentiality of mitochondrial DNA**. Based on whether mtDNA is essential or not, yeast species have been classified as either "petite-negative" or "petite-positive"[26]. *S. pombe* is a petite-negative yeast that cannot survive without mtDNA[27]. It has been reported that certain nuclear mutations can convert *S. pombe* into a petite-positive yeast, but genes underlying these mutations remain unidentified[27,28].

All seven query genes that function in mitochondrial translation are bypassable and share a common set of 12 BOE suppressors (Fig. 2 and Supplementary Fig. 3a). Because a failure to express mtDNA-encoded genes is equivalent, in consequence, to mtDNA loss, we hypothesized that these suppressors may also render mtDNA dispensable. Indeed, mtDNA loss can be readily induced in strains carrying any one of these suppressors but not in a wild-type control strain (Supplementary Fig. 3b). Thus, BOE analysis led to the identification of genes whose alteration can convert *S. pombe* into a petite-positive species.

One mtDNA-bypassing C-BOE suppressor, *atp3-R282C*, is a mutation in the gene encoding the gamma subunit of mitochondrial F1-ATPase. Mutations in this gene have been shown to render *Kluyveromyces lactis*, another petite-negative yeast, and the protist *Trypanosoma brucei*, tolerant of mtDNA loss[29,30]. Such mutations are believed to increase the ATP hydrolysis capacity of F1-ATPase and thereby allow the mitochondrial inner membrane potential to be maintained in the absence of mtDNA.

Another mtDNA-bypassing C-BOE suppressor, *mts4-S412F*, is a mutation in the gene encoding a 19S proteasome subunit. Probably not by coincidence, 6 of the 10 mtDNA-bypassing OP-BOE suppressor genes encode either proteasome subunits or proteasome-associated proteins (Supplementary Fig. 3a), suggesting that proteasome alteration is a common mechanism of mtDNA bypass in *S. pombe*, even though this mode of mtDNA bypass has not been reported before in any petite-negative species. Thus, through BOE analysis we uncovered a previous unknown link between the proteasome and mtDNA dispensability.

Interestingly, overexpression of Dicer (Dcr1), a ribonuclease known to be a limiting factor of the *S. pombe* RNAi pathway[31], can also bypass mtDNA (Supplementary Fig. 3b), suggesting that upregulating the RNAi pathway promotes cell survival against mtDNA loss.

**Protein complex subunits tend to share bypassability**. The shared bypassability of mitochondrial translation genes suggested that bypassability is associated with particular functional modules, and the most common functional modules are protein complexes. Upon examining protein complexes containing at least two subunits encoded by chrII-L query genes, we found that the constituent subunits of a given essential protein complex indeed tend to be either all bypassable or all non-bypassable (Fig. 4a and Supplementary Data 3). In three follow-up analyses, we found that when a complex contains one known bypassable subunit, the other subunits turned out to be bypassable as well (Supplementary Figs. 3c-e). Thus, subunits belonging to the same protein complex tend to share bypassability and complex membership can be used to predict gene bypassability if the bypassability of one complex member is known.

To see whether protein complex bypassability can be predicted a priori, we broadly surveyed complex features and identified 9 features significantly correlated with complex bypassability (Supplementary Figs. 4a, b). Using these features to perform hierarchical clustering of 127 essential protein complexes in *S. pombe* (Fig. 4b and Supplementary Data 4), we found that all five of the bypassable complexes (names in dark blue in Fig. 4b) defined by our BOE analysis fell into a cluster containing 17 complexes (blue branches in the dendrogram in Fig. 4b). We noted that this same cluster includes five additional complexes (names in light blue in Fig. 4b) whose bypassability is known or expected. Therefore, we predicted that the other complexes in this cluster, including the THO complex (name in black in Fig. 4b), are likely to be bypassable. To test this prediction, we performed a T-BOE screen for suppressors of *tho2*, which encodes an essential subunit of the THO complex, and found that *tho2* is indeed bypassable and that its deletion mutant can be rescued by deleting *git1*, *cyr1*, or *pka1*, three genes acting in the cAMP-protein kinase A signaling pathway (Supplementary Fig. 4c).

**Differences between complex subunits are revealed**. In cases where subunits of the same protein complex do not share bypassability or BOE suppressors, new insights on the functional differences between subunits can be gained. Upon surveying the protein complexes with two or more subunits analyzed in our study, we identified three types of non-uniformity between subunits of the same complex (Supplementary Fig. 5).

First, there were four cases of "mixed complexes", with each comprised of both bypassable and non-bypassable essential subunits (Supplementary Fig. 5a). There is evidence from the literature suggesting that three of these four complexes are assembled from subunits that have either non-overlapping functions or different functional importance[32–34]. For the fourth

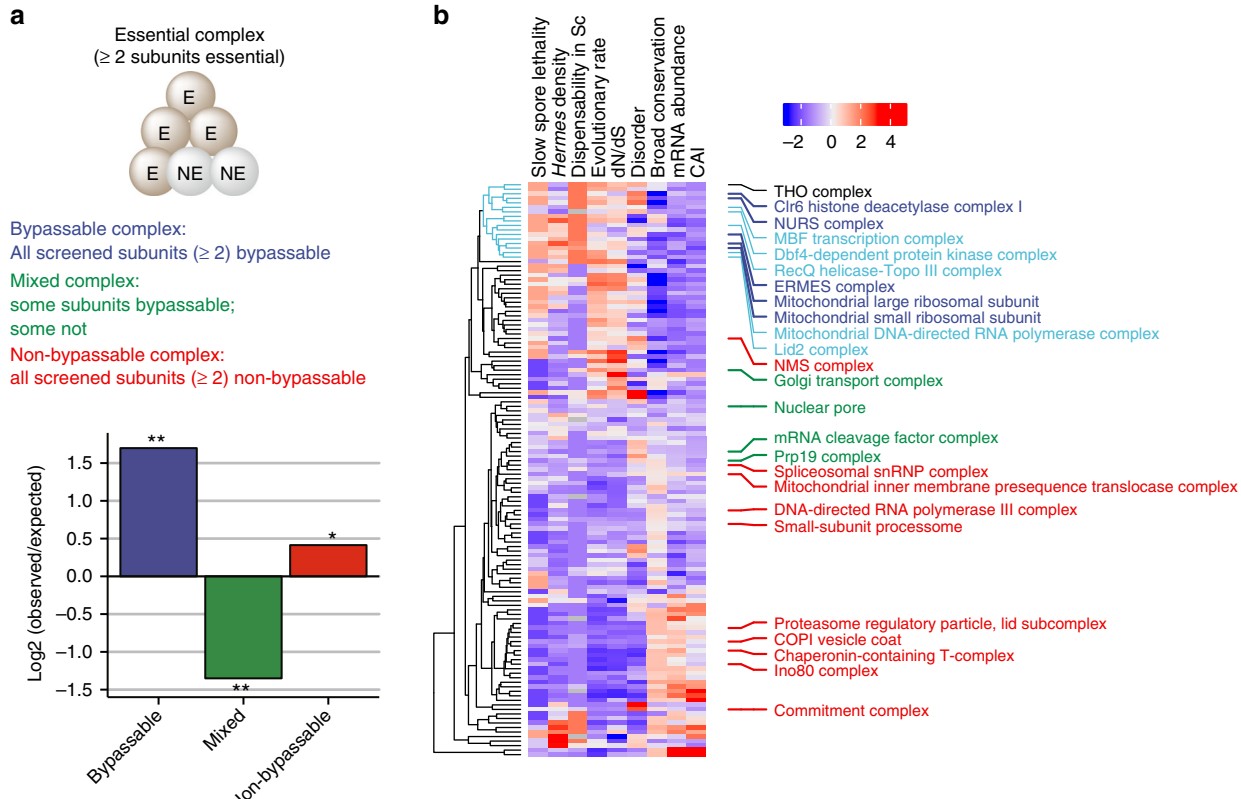

**Fig. 4** Bypassability tends to be shared by subunits belonging to the same protein complex. **a** Subunits of the same protein complex tend to be either all bypassable or all non-bypassable. The observed numbers of the three types of complexes were compared to the expected numbers computed as the arithmetic mean of 1000 random permutations. ** indicates $P < 0.01$, and * indicates $P < 0.05$. $P$ values were calculated from the permutations. **b** Hierarchical clustering of essential protein complexes based on nine features that have the best predictive power for complex bypassability. The complexes named in the figure include the bypassable complexes (dark blue), mixed complexes (green), and non-bypassable complexes (red) classified based on the criteria shown in **a**. The cluster that all bypassable complexes fall into is highlighted in blue in the dendrogram. Within this cluster, the names of five additional complexes known or expected to be bypassable are in light blue, and the name of the THO complex is in black. The complexes are listed in Supplementary Data 4 in the same order as here

complex, the Prp19 complex, our findings that one of its subunits, Cwf15, is bypassable, whereas two other subunits (Cwf7 and Prp5) are non-bypassable, reveal previously unappreciated functional difference between its subunits.

Second, bypassable subunits belonging to the same complex occasionally do not share the same BOE suppressors. The histone deacetylase Clr6 occurs in two complexes (Clr6 complex I and Clr6 complex II)[35,36]. Clr6 and four complex I-specific subunits (Pst1, Pst3, Sds3, and Rxt3) are essential, whereas none of the complex II-specific subunits are essential. Thus, it has been assumed that the essential function of Clr6 is that of complex I[35]. Clr6 and the complex I-specific subunit Pst3 were shown by our BOE analysis to be bypassable, but they do not share any BOE suppressors (Fig. 2). *clr6* deletion can be rescued by mutations disrupting either the AMPK complex or the CCR4-NOT complex, while *pst3* deletion can be rescued by deleting Clr6 complex II genes (*pst2* or *cph2*). Our follow-up analyses showed that the other three complex I-specific essential subunits, Pst1, Sds3, and Rxt3, can also be bypassed by *pst2* deletion (Supplementary Fig. 5b).

These results suggest that the essentiality of Clr6 complex I is at least partly due to its role in antagonizing complex II. Interestingly, simultaneously deleting all three paralogous *pst* genes (*pst1*, *pst2*, and *pst3*) resulted in lethality (Supplementary Fig. 5b), suggesting that besides their functions in counteracting complex II, the two complex I components Pst1 and Pst3 redundantly contribute to a growth-promoting function. This

function is probably Clr6-dependent, as the lethality of *pst1 pst2 pst3* triple deletion can be rescued by a BOE suppressor of *clr6* (Supplementary Fig. 5b). Thus, a more intricate than expected relationship among the components of Clr6 complexes was revealed by our BOE analysis. In another example, we found that two of the three essential subunits of the NURS complex[37], Pir2 and Red5, are bypassable but share no common suppressor (Fig. 2). Follow-up analysis showed that the third essential subunit of this complex, Mtl1, is also bypassable by a suppressor of *red5* (Supplementary Fig. 5c), indicating that Mtl1 and Red5 may have a closer relationship with each other than with Pir2.

Third, a unidirectional suppression relationship between subunits was observed for the ERMES complex, which acts as an ER–mitochondrion tether. Genes encoding the four ERMES subunits, *mmm1*, *mdm10*, *mdm12*, and *mdm34*, are all bypassable essential genes (Fig. 2 and Supplementary Fig. 5d). Interestingly, overexpression of *mmm1* can suppress the lethality of *mdm34Δ*, *mdm12Δ*, and *mdm34Δ mdm12Δ* double deletion, but not *mdm10Δ* (Supplementary Fig. 5d). Overexpression of *mdm10*, *mdm12*, or *mdm34* cannot suppress the deletion of any other ERMES genes (Supplementary Fig. 5d). These results suggest that the four ERMES subunits are not equally important, with Mdm34 and Mdm12 playing a more peripheral role.

**Essentiality-bypassing mechanisms**. We next investigated the mechanisms of essentiality bypass. Redundancy between paralogous genes is a well-known mechanism underlying synthetic

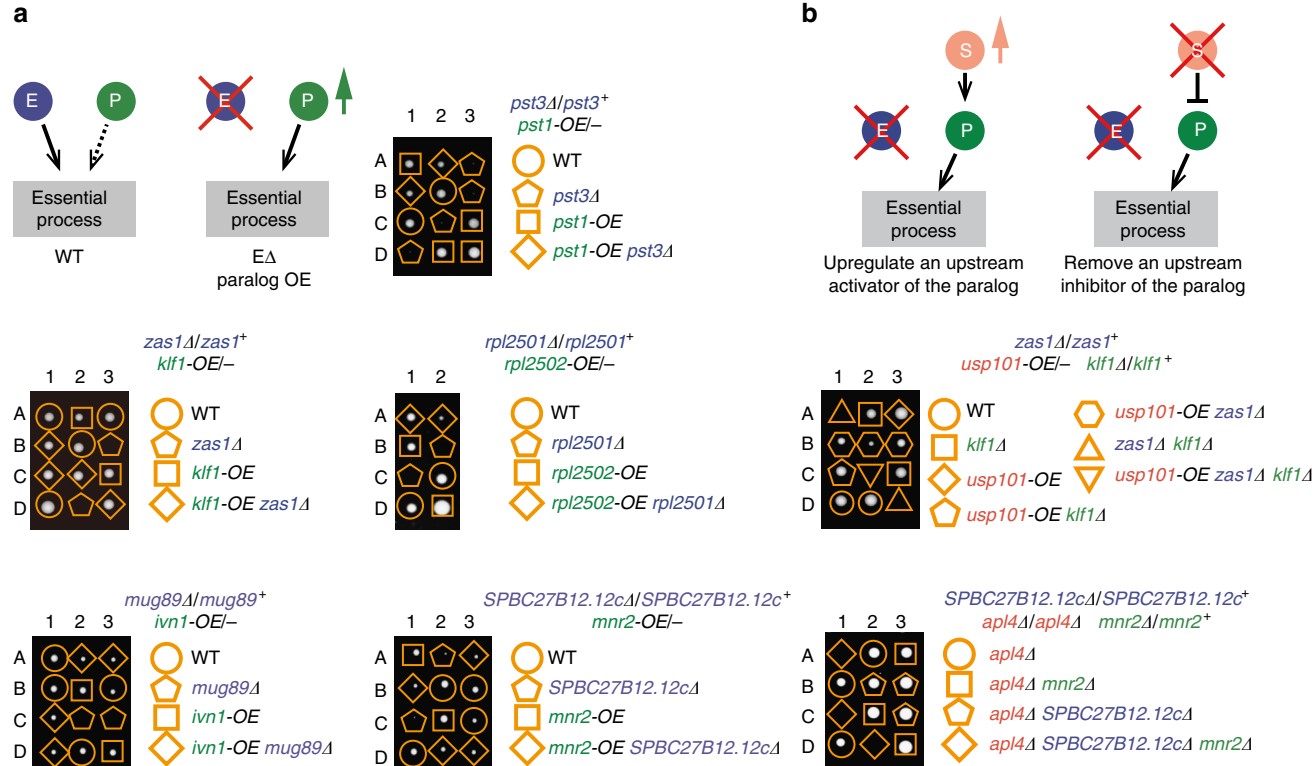

**Fig. 5** Activating a dormant redundancy between paralogs is one mechanism of essentiality bypass. **a** Schematic illustrating how an essential gene ('E' in blue) can be rendered dispensable by the overexpression (OE) of its paralog ("P" in green), and tetrad dissection showing that the essentiality of five query essential genes can be bypassed by overexpressing their respective paralogs. **b** Two hypothetical scenarios of paralog-dependent bypass achieved by altering a non-paralogous suppressor gene ("S" in orange) and two examples of paralog-dependent bypass

lethality[38,39]. We surmised that essentiality bypass may result from an inverse mechanism—the activation of a dormant redundant paralog of an essential gene. To determine the extent to which upregulating the expression of a paralog can achieve a bypass, we directly tested whether the nine query genes that have paralogs can be bypassed by paralog overexpression. For five of them, their essentiality was indeed bypassed via overexpressing their paralogs (Fig. 5a and Supplementary Table 1). Among these five query genes that can be bypassed by paralog overexpression, four can also be bypassed by suppressors that alter non-paralogous genes (Fig. 2 and Supplementary Data 2). Triple mutant analysis revealed that at least some of these non-paralog suppressors act in a paralog-dependent manner (Fig. 5b), suggesting that in the wild-type situation, a non-paralog suppressor gene may play a sensitizing role by keeping the paralog in the dormant state. To what extent this model can explain the suppressor-paralog relationships in the two cases shown in Fig. 5b awaits further analysis. It is possible that some paralog-dependent suppressors may not act through activating the paralog, but rather through reducing the need for the function of the essential gene, thus allowing a weak backup activity to become sufficient for supporting viability. An example that mechanistically conforms to the model of activating a dormant redundant paralog will be given below.

Between-paralog dormant redundancy can only explain a small fraction of the BOE interactions. Through surveying BOE interactions that can be explained at the molecular level (including those reported in the literature), we classified the mechanisms of explainable BOE interactions in fission yeast into three types: bypass by avoiding toxic intermediate formation, bypass by downstream compensation, and bypass by parallel compensation (Supplementary Fig. 6). Based on these

classifications, the scenarios that can give rise to essentiality bypass include: a toxicity-generating step in an otherwise non-essential process (type 1 mechanism), multi-step regulation (type 2), as well as dormant redundancy and counterbalancing activities (type 3). We note that most of the BOE interactions uncovered in this study cannot yet be classified into one of these mechanism types, owing to a shortage of molecular-level evidence on the query-suppressor relationships. It is possible that a substantial fraction of BOE interactions occur through other types of mechanisms, including mechanisms seemingly unrelated to the functions of the query genes[8,40].

**BOE interactions connect functionally related genes.** BOE interactions connect genes either directly (a query gene and a suppressor gene form an "interacting pair") or indirectly (two suppressor genes that suppress the same query gene deletion form an "interactor-sharing pair") (Fig. 6a). We asked to what extent genes linked by BOE interactions are functionally related. Analysis of our BOE data indicated that, for BOE interacting pairs, a query gene and its BOE suppressor gene have a high probability of sharing the same biological process GO slim term (Fig. 6b), of encoding protein products that localize to the same subcellular compartment(s) (Fig. 6c), and of encoding protein products that physically interact with each other (Fig. 6d). The enrichment ratios are notably higher for BOE interacting pairs than for genetic interacting pairs identified by global mapping in fission yeast using null alleles of non-essential genes and hypomorphic alleles of essential genes[41], even after selecting only the top 10% of the strongest interactions from the global mapping data. Similar enrichment patterns were observed for interactor-sharing pairs (Fig. 6e–g). Interestingly, interactor-sharing pairs are enriched

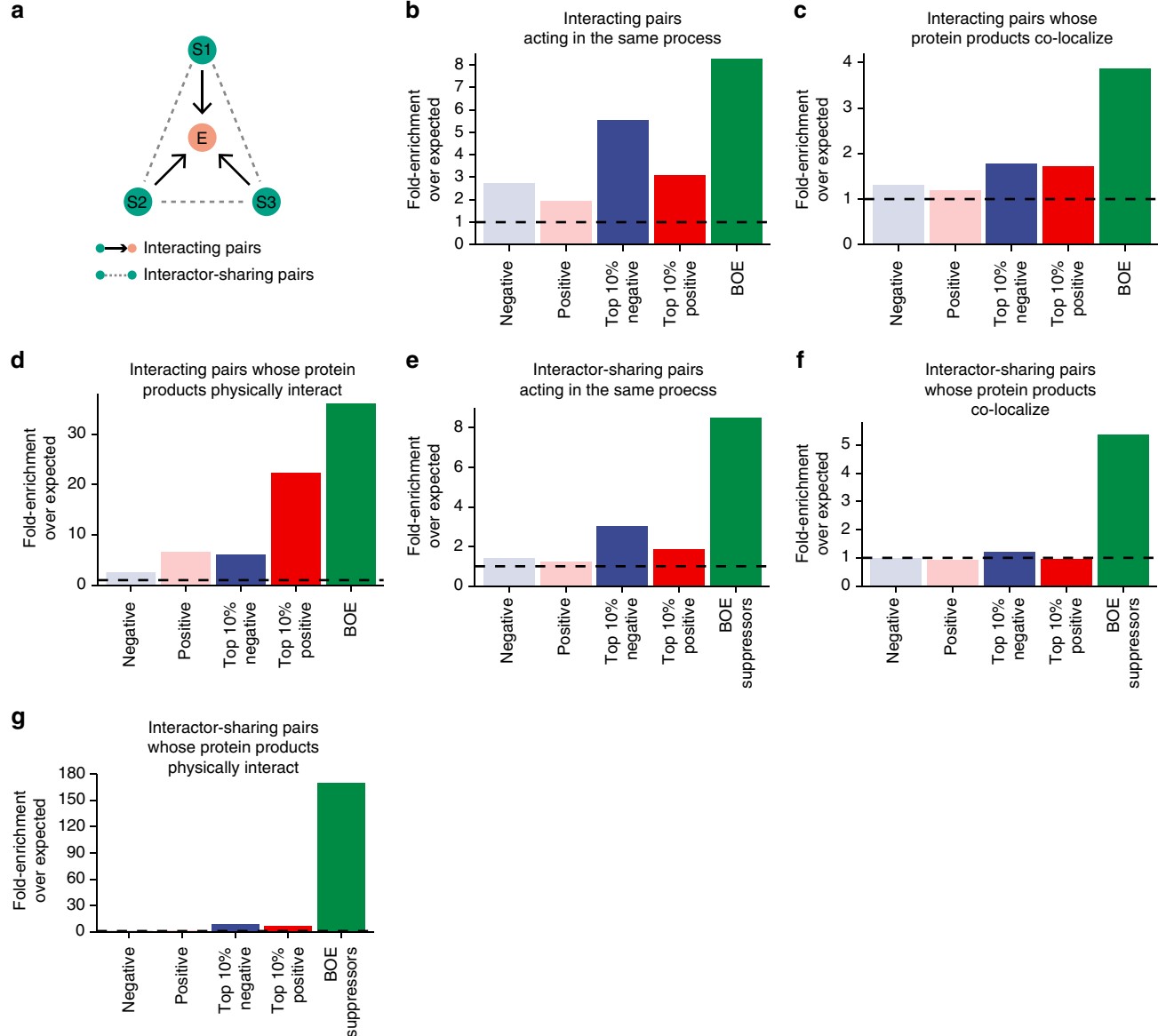

**Fig. 6** BOE interactions are strongly correlated with functional relatedness. **a** Schematic depicting the two types of gene pairs for which the enrichment of functionally related genes within pairs were analyzed. **b**–**d** The numbers of observed interacting gene pairs in which **b** both genes are associated with the same biological process GO slim term, **c** the protein products of both genes are located in the same subcellular location, or **d** the protein products of the two genes physically interact, were compared with the expected numbers from all possible pairs generated by permutation. **e**–**g** The numbers of observed interactor-sharing gene pairs in which **e** both genes are associated with the same biological process GO slim term, **f** the protein products of both genes are located in the same subcellular location, or **g** the protein products of the two genes physically interact, were compared with the expected numbers from all possible pairs

with physically interacting proteins to a considerably greater extent than BOE interacting pairs, probably because suppressor genes for the same query often include genes encoding multiple subunits of the same protein complex. In sum, we find that BOE interactions are stronger predictors of functional relatedness than genetic interactions mapped using non-lethal mutations.

**Inferring gene functions with BOE analysis**. We explored the utility of BOE analysis for predicting gene function by conducting three follow-up studies. In the first, during the pilot phase of this project, we performed a T-BOE screen of the cohesin gene *rad21*, which is known to be bypassable via upregulating its meiosis-specific paralog *rec8*[42]. The top screen hit was *erh1*, a gene whose function was unknown at the time (Supplementary Fig. 7a). In the

same screen, we also isolated mutations affecting *mmi1*, a known repressor of meiotic genes that contain DSR (determinant of selective removal) motifs, including *rec8*[43], raising the possibility that *erh1* and *mmi1* function in the same pathway (Fig. 7a). A series of follow-up analyses found that Erh1 physically interacts with Mmi1 (Supplementary Figs. 7b, c), co-localizes with Mmi1 (Supplementary Figs. 7d, e), homo-oligomerizes (Supplementary Fig. 7 f), and mediates Mmi1–Mmi1 interactions (Supplementary Fig. 7 g). Using a DSR-containing reporter, we found that the repression defect of *erh1Δ* can be rescued by artificial dimerization of Mmi1 (Fig. 7b). RNA-seq analysis showed that the de-repression of endogenous genes in *erh1Δ* was also reversed by artificial Mmi1 dimerization (Fig. 7c and Supplementary Data 5). Thus, BOE analysis generated the initial clue that Erh1 may act together with Mmi1, and our follow-up analyses established that

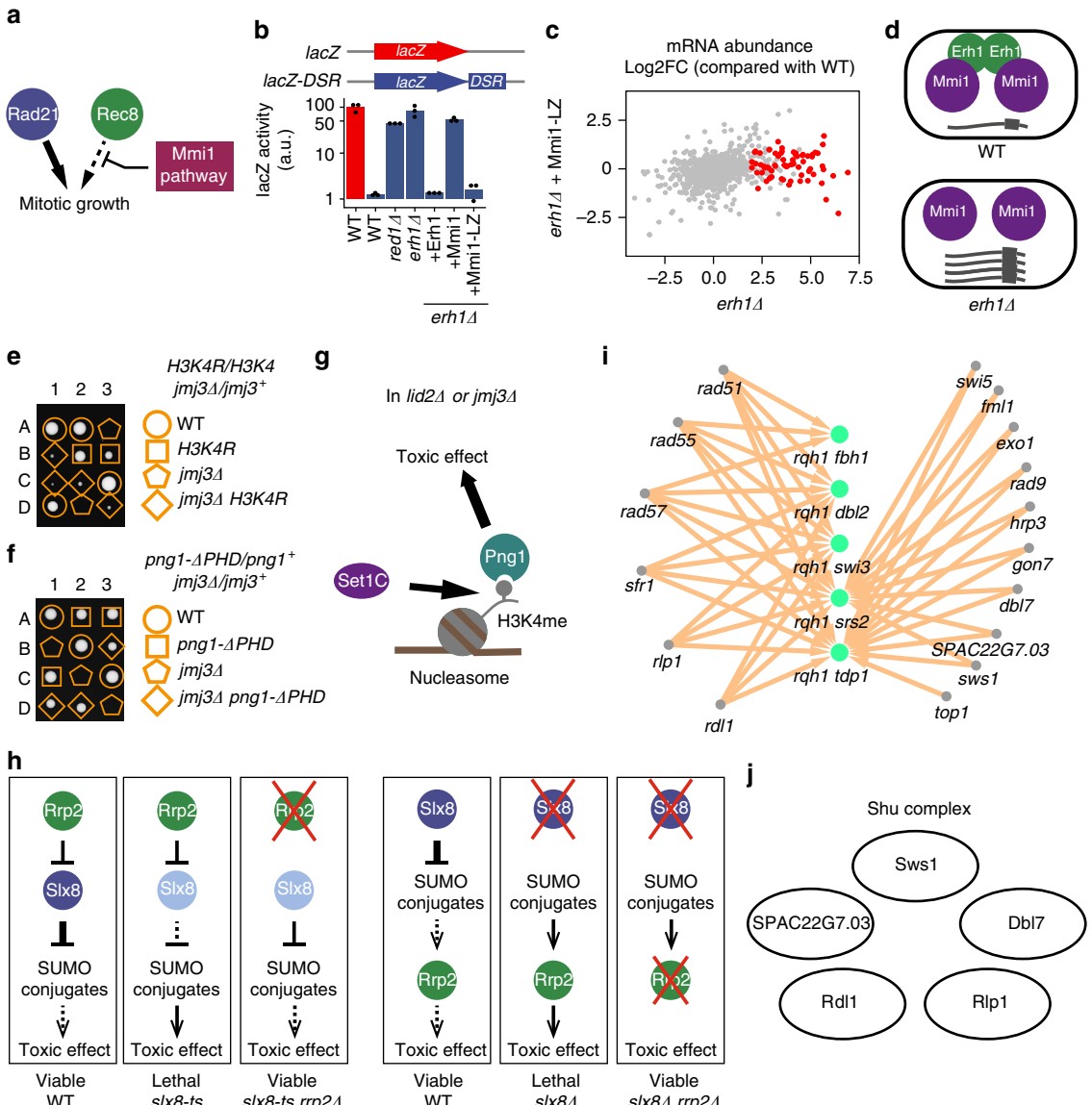

**Fig. 7** Examples demonstrating the power of BOE analysis to infer gene functions. **a** Schematic showing the relationships between *rad21*, its paralog *rec8*, and the Mmi1 pathway. **b** Mmi1 fused with a homodimeric leucine zipper (Mmi1-LZ) rescued the gene repression defect of *erh1Δ*. DSR-fused *lacZ* gene (*lacZ-DSR*) was used as a reporter. *lacZ* expressed in WT was used as a normalization control. **c** A scatter plot showing log2-transformed fold changes (Log2FC) of mRNA abundance relative to WT in *erh1Δ* and *erh1Δ* expressing Mmi1-LZ. mRNAs upregulated in *erh1Δ* are highlighted in red. **d** A model of how Erh1 functions. **e** *jmj3* can be bypassed by the K4R mutation in histone H3. **f** *jmj3* can be bypassed by the PHD domain truncation of *png1*. **g** A model depicting the Set1C-H3K4me-Png1 axis that causes the lethality of *lid2Δ* and *jmj3Δ*. **h** A model explaining how *rrp2Δ* can suppress *slx8-ts* (left) and a model explaining how *rrp2Δ* can suppress *slx8Δ* (right). **i** BOE analysis identified 16 genes (gray nodes), when individually deleted, can suppress the lethality of at least one of five synthetic-lethal double deletion mutants (green nodes). **j** A diagram depicting the fission yeast Shu complex, including the two subunits newly identified in this study

the main role of Erh1 is to oligomerize Mmi1 (Fig. 7d). Our results are consistent with and complementary to recently published studies on Erh1[44,45]. Triple mutant analysis (Supplementary Fig. 7 h), together with the mechanistic understanding on Erh1, firmly established that the bypass of *rad21* by *erh1Δ* is through the de-repression of *rec8*, a dormant redundant paralog of *rad21*.

The second study illustrates the use of iterative BOE analysis. We initially found that *lid2* and *jmj3*, which respectively encode a histone H3K4 demethylase and its binding partner[46,47], can both be bypassed by disrupting the Set1/COMPASS H3K4 methylase complex (Set1C) (Fig. 2 and Supplementary Fig. 3d). Furthermore, H3K4R mutation can also suppress the lethality of *jmj3Δ*

(Fig. 7e). Thus, when the Lid2-Jmj3 complex is defective, Set1C-mediated H3K4 methylation becomes lethally toxic to the cell. To identify factors acting downstream of H3K4 methylation, we performed another round of BOE analysis, specifically screening for *jmj3* C-BOE suppressors that do not alter the level of H3K4 methylation. We isolated a premature stop mutation (W269*) in the PHD domain of *png1*, which encodes a subunit of the NuA4 histone acetylase complex[36]. A *png1* truncation allele lacking the PHD domain, but not *png1Δ*, could bypass *jmj3* (Fig. 7f and Supplementary Fig. 7i). We found that the PHD domain mediates Png1 binding to K4-methylated H3 and specifically blocking this binding was sufficient to rescue *jmj3Δ* (Supplementary Figs. 7j-l). Together, these results suggest that Png1 is the downstream

reader protein underlying the toxic effect of H3K4 methylation when the Lid2-Jmj3 complex is compromised (Fig. 7g).

Both of the above two studies capitalized on the functional relatedness of BOE suppressor genes that suppress the same query gene deletion. In the next example, we show that BOE interactions can clarify the relationships between a query gene and its suppressor genes. Our systematic BOE analysis identified many suppressors that can bypass *slx8*, which encodes the catalytic subunit of the SUMO-targeted ubiquitin ligase (STUbL) (Fig. 2). Among them, *rrp2Δ* was previously shown to be a suppressor of an *slx8* temperature-sensitive (ts) mutant[48]. Because Rrp2 is a STUbL inhibitor[49], the suppression of *slx8-ts* by *rrp2Δ* can be attributed to the loss of STUbL inhibition (Fig. 7h, left side). Note that this model does not predict a bypass of *slx8* by *rrp2Δ*. Our bypass result demonstrated that the suppression effect of *rrp2Δ* does not require any residual STUbL activity, a conclusion that cannot be drawn based on the suppression of a ts mutant. Our follow-up analyses showed that Rrp2 may mediate the toxic effect of the SUMO conjugates that accumulate in *slx8Δ* (Supplementary Figs. 7m-q and Fig. 7h, right side).

**Applying BOE analysis to conditionally essential genes.** Besides genes that are essential in the wild-type background, BOE analysis can also be applied to genes that are conditionally essential (e.g., non-essential genes that become essential in the presence of a synthetic-lethal mutation). As a proof of principle, we performed T-BOE screens on five synthetic-lethal double mutants that all include the deletion of the RecQ helicase gene *rqh1*. Among these double mutants, *rqh1Δ srs2Δ* is known to be suppressed by deleting any one of eight early homologous recombination (HR) genes including *rad51*, *rad55*, *rad57*, *sfr1*, *swi5*, *sws1*, *rlp1*, and *rdl1*, the latter three of which encode the three known subunits of the fission yeast Shu complex[50,51]. From among the candidate suppressor genes uncovered by our screens, we chose 16 genes to perform verification using their deletions, and thereby confirmed a total of 55 suppression interactions, including all of the previously known *rqh1Δ srs2Δ* suppressors (Fig. 7i). Follow-up analysis on *dbl7* and *SPAC22G7.03*, two suppressor genes without any known role in HR, showed that they encode previously unknown components of the Shu complex (Supplementary Figs. 7r–t and Fig. 7j). The discovery of new factors acting in HR, one of the most intensively studied processes in fission yeast, demonstrates the power of applying BOE analysis to conditionally essential genes, which should include a majority of the non-essential genes[52].

## Discussion

This study unveils a surprisingly large extent of "hidden dispensability" among essential genes and demonstrates that essentiality bypass by monogenic suppressors is a previously under-appreciated mechanism of essentiality change. Importantly, our discoveries shed new light on how gene essentiality may be shaped by systems-level properties. Analogous to the situation wherein synthetic lethality reveals a systems-level property called "genetic robustness"[39], bypassable essentiality reveals a form of "anti-robustness"—hypersensitivity to mutations[53]. We propose that, like robustness, anti-robustness is also a ubiquitous property of biological systems and it has a large but under-recognized impact on genotype-phenotype relationships.

Gene bypassability revealed by BOE analysis is related conceptually to the quantitative definitions of gene essentiality proposed by Rancati et al.[54]. Rancati et al.'s definitions of "low essentiality genes" and "high essentiality genes" based on whether compensatory mutations occur at high frequency or at low frequency, together correspond to bypassable essential genes defined in this study. In the future, more comprehensive BOE analyses may allow informative comparisons between bypassable genes with many BOE suppressors and the ones with few BOE suppressors. Rancati et al.'s definition of "complete essentiality genes" based on the lack of compensatory mutations corresponds to non-bypassable essential genes defined in this study. We note that despite our efforts to maximize the coverage of mutational space, it remains possible that some of the non-bypassable genes defined here are actually bypassable by rare monogenic suppressors missed by our BOE analysis, or can only be rendered dispensable by simultaneously mutating multiple genes.

We demonstrate here that BOE analysis is a powerful method that can be applied in both systematic and gene-focused studies, and can be used to investigate both essential genes and non-essential genes. Because BOE suppressors can be isolated by tight growth-based positive selection, BOE analysis can maximally take advantage of the power of random mutagenesis-based forward genetic screening, a methodology that can explore a much larger suppressor mutation space than approaches based on deletion libraries. For 26% (10/38) of the bypassable essential genes identified in this study, the only type of suppressors that we uncovered were point mutations; these bypassable genes would have been missed without the chemical mutagenesis approach. *PB* transposon-based screening also uncovered suppressors that could be verified using insertion alleles but not deletion alleles (Supplementary Data 2). Moreover, our results highlight the fact that even null allele suppressors can be absent in a haploid deletion library of non-essential protein-coding genes—our identified suppressors included null alleles of an essential gene (*fap7Δ*, which can bypass *wdr74*) and an RNA gene (*snR101Δ*, which can bypass *cwf15*).

The prevalence and principles of essentiality bypass revealed by our study should facilitate future research about gene essentiality and genetic network wiring, and may aid the efforts of synthetic biologists in building minimal and designer genomes. The methodology of BOE analysis, with proper modifications, should be applicable to many other organisms including human cells.

The insights gained on bypassable essentiality are also of value to medicine. On the one hand, in drug discovery projects aiming to inhibit essential genes, drug targets can be prioritized according to gene bypassability—this could pre-emptively avoid/minimize drug resistance complications. For example, in the development of drugs that act against eukaryotic pathogens such as fungi and protozoa, an often-mentioned guiding principle is to select drug target genes that are uniquely important to the pathogen but not the host. Our finding that genes essential in one eukaryotic species but not another tend to be bypassable raises a cautionary note to the use of such a rationale. On the other hand, in the many situations where a defective essential gene is the underlying cause of a disease, and particularly when chemical-based activation is difficult to achieve, the identification of loss-of-function bypass suppressors would define entirely new targets for which small molecule inhibitors can be developed as drugs.

## Methods

**Plasmids and strains.** Plasmids and *S. pombe* strains were constructed using standard practices. Strains used in the follow-up analyses in this study are listed in Supplementary Data 6. All plasmids and fission yeast strains generated in this study are available upon request.

**Compilation of a list of essential genes on chrII-L.** Essential genes on chrII-L were selected according to gene dispensability information in column I of Supplementary Table 1 in Hayles et al. 2013[23,55]. Based on two small-scale studies[56,57], we added *SPBC27B12.08* (*sip1*) as an essential gene, and excluded *SPBC1348.06c*, *SPBC359.02* (*alr2*), and *SPBC1198.02* (*dea2*).

**Rescuing plasmid construction.** The following four versions of rescuing plasmids were used in this study. (1) pPC96-LEU2-Ptub1-ORF-CFP-Flag-His6 derived from the Yoshida ORFeome plasmid library. First, the Gateway destination vector pPC96-LEU2-Ptub1-ccdB-CFH (pDB790) vector was constructed by inserting the marker gene *S. cerevisiae LEU2* and the Ptub1-ccdB-CFH fragment from pHIS3K-CFH21c into the *ade6+*- and *TK*-containing vector pPC96, which is also known as pFS119 or pNR210[58]. pPC96-LEU2-Ptub1-ORF-CFH plasmids were constructed by transferring the essential gene ORFs from the entry clones of the Yoshida ORFeome plasmid library into this destination vector using the Gateway LR recombination reaction[59]. (2) pPC96-LEU2-Ptub1-ORF-11aa derived from the Yoshida ORFeome plasmid library. The Gateway destination vector pPC96-LEU2-Ptub1-ccdB (pDB1801) was constructed by removing the CFH tag from pPC96-LEU2-Ptub1-ccdB-CFH. Plasmids obtained using this destination vector, the Yoshida ORFeome entry clones, and the Gateway LR reaction are called pPC96-LEU2-Ptub1-ORF-11aa plasmids, as the essential gene ORFs are fused in the C-termini with 11 extra amino acid translated from the attB sequence. (3) pPC96-LEU2-Ptub1-ORF-TAA generated independently of the ORFeome library. Essential gene ORFs were amplified from genomic DNA using PCR primers that render all ORFs ending with TAA as the stop codon, and inserted into a modified pPC96-LEU2-Ptub1 vector (pDB1819) using the In-Fusion reaction. (4) pPC96-ORF. For a small number of essential genes analyzed in the pilot phase of this study, the essential gene ORF together with the upstream region were amplified from genomic DNA and inserted into pPC96. The resultant plasmids do not have the *LEU2* marker.

**BOE query strain construction.** Two strategies were used to construct the query strains for BOE screens.

In the first strategy, an *h-/h- ade6-M210/ade6-M210 arg6::PB[ura4+]/arg6::PB[ura4+] ura4-D18/ura4-D18 leu1-32/leu1-32 his3-D1/his3-D1* diploid strain (DY5063) was mated with individual *kanMX* (G418-resistance)-marked heterozygous deletion diploid strain in the background of *h + /h + ade6-M210/ade6-M216 ura4-D18/ura4-D18 leu1-32/leu1-32* (Bioneer)[55]. After removing unmated parental cells using glusulase treatment, spores were allowed to germinate on YES plates. Then we replica-plated the YES plates to the low-adenine YE + G418 plates and PMG plates lacking uracil and arginine to select diploid progenies that are homozygous for *ade6-M210* (red colony color on YE + G418) and heterozygous for both the essential gene deletion and *arg6::PB[ura4+]* (ability to grow on YE + G418 and PMG without uracil and arginine, respectively). Single colonies were restreaked and tested for their ability to undergo meiosis (*h + /h−*). Then the corresponding rescuing plasmid was introduced into the cells by transformation. For rescuing plasmids with both *LEU2* and *ade6+* markers, transformants were selected on PMG plates lacking both leucine and adenine. For plasmids with *ade6+* only, the selection was conducted with PMG plates without adenine. Spores derived from three independent transformants were collected from the selective PMG plates directly because meiosis can happen on PMG plates. After glusulase digestion, about 10,000 spores were spread on PMG plates lacking uracil, leucine, and adenine. For the *ade6+*-only plasmids, leucine was supplied. After colony formation, the PMG plates were replica-plated to the low-adenine YE + G418 plates. White colonies, which were deletion mutant haploids that needed the *ade6+* marker-containing plasmid to survive, were selected. Red colonies, which presumably were diploids, were avoided. Then we restreaked the colonies to YES plates and confirmed the episomal state of the plasmid by inspecting under the microscope after incubation for at least 20 h. The high-frequency appearance of microcolonies or un-divided cells indicated that the plasmid remained episomal and unstable, whereas universally healthy growing colonies indicated that the plasmid had integrated into the genome and become stable.

In the second strategy, the rescuing plasmid was introduced into DY11115 (*h-leu1-32 ura4-D18 ade6-M210 arg6::PB[ura4+]*) by transformation and transformants were selected on PMG plates without leucine and adenine. The corresponding essential gene on the chromosome was deleted using PCR-based gene targeting. Upon transforming a *kanMX* marker-containing PCR product into cells containing the rescuing plasmid, the cells were recovered on the selective PMG plates for 2 days and then replica-plated onto low-adenine YE + G418 plates. White colonies were chosen and the episomal state of the rescuing plasmid was validated using the same procedure as in the first strategy.

**C-BOE screens using the chemical mutagen MNNG.** Cells of query strains pre-grown in the YES medium were harvested and washed with TM buffer (50 mM Tris-maleic acid, pH 5.8). Five OD600 units of cells ($1 \times 10^8$ cells) were resuspended in 300 µL of TM buffer and 100 µL of 2 mg/ml MNNG stock solution (in TM buffer) was added. After incubation for 60 min at the room temperature, cells were spun down and washed with TM buffer twice. Such an MNNG treatment resulted in around 10% cell survival. Then the cells were resuspended in 400 µL of water and spotted on YES plates with 20 µL per spot. After 3–4 days incubation at 30 °C, each master YES plate was replica-plated to three YE + G418 + FUdR plates, which were then incubated at 25 °C, 30 °C, and 33 °C, respectively. Red or pink colonies on YE + G418 + FUdR plates were restreaked to single colonies and then, colony PCR was performed to confirm the absence of the essential gene ORF. For each query gene, at least two independent C-BOE screens were conducted.

**Mapping the C-BOE suppressors using BSA-seq.** Strains isolated from C-BOE screens were backcrossed to a wild-type strain. Through either tetrad dissection or random spore analysis, we obtained viable G418-resistant progenies of the cross and pooled them together. Genomic DNA was extracted from the pool and genomic re-sequencing libraries were constructed using the NEBNext DNA Library Prep Master Mix Set for Illumina kit. Single-read sequencing was performed using Illumina HiSeq 2000 or 2500.

Read mapping to the reference genome (Schizosaccharomyces_pombe. ASM294v1.18.dna.toplevel.fa.gz, last modified 29 April 2013) was performed using BWA-MEM version 0.7.7. After duplicate removal, single-nucleotide polymorphisms (SNPs) were called by SAMtools version 0.1.18. SNPs with the quality score less than 100 and supported by less than two reads were discarded. In addition, we removed SNPs present in a manually compiled background SNP list and those linked to the query essential gene (i.e., with a distance of less than 500 kb). For each of the remaining SNPs, we computed the percentage of reads supporting the reference allele using the software bam-readcount and predicted the mutation effect using the software Coovar version 0.07 with the annotation file Schizosaccharomyces_pombe.ASM294v1.18.gtf. A candidate SNP suppressor mutation should meet two criteria: (1) the SNP should be either a missense mutation, a stop codon-gain or -loss mutation, or a splicing site mutation, and (2) the percentage of reference allele-matched reads should be less than 10%.

**T-BOE screens using the *PB* transposon.** We created a *pha2*-targeted integration plasmid, pPHA2H (pDB733), by replacing the partial *his3* fragment in pHIS3H[60] with a partial *pha2* fragment harboring an engineered NotI site. pPHA2H-Pnmt1-PBase (pDB822) was constructed by inserting Pnmt1-PBase fragment into pPHA2H cut by SphI and BamHI, and integrated into the query strains.

For each query gene, we conducted at least two independent screens. Log-phase cells grown in YES were washed with water three times and spotted on thiamine-free PMG + Arg−Ura plates to induce PBase expression. Each spot contained about 0.004 OD unit of cells and twenty spots were used for each screen. PBase-mediated transposition of *PB[ura4 + ]* from the *arg6* locus to other sites restored a functional *arg6 +* and rendered cells arginine prototrophic[61]. After 2–3 days, cells were washed from PMG + Arg−Ura plates and about two OD600 units of cells ($4 \times 10^7$ cells) were plated onto PMG−Arg + Ura + thiamine plates to shut off PBase expression and enrich transposition events by selecting Arg + cells. Based on a typical transposition efficiency of 2%, there should be $8 \times 10^5$ Arg + cells harboring transpositions. After 3–4 days, cells on the PMG−Arg + Ura + thiamine plates were collected and two OD600 units of cells transferred to YE + G418 + FUdR plates. The YE + G418 + FUdR plates were incubated at 30 °C until colonies appeared. Red colonies were picked and pooled together.

**Identifying *PB* insertion sites by Junction-seq.** *PB* insertions in the pools of FUdR-resistant and red-colored colonies were profiled using Illumina sequencing[61]. Briefly, a primer extension reaction with a primer (oligo-128, Supplementary Table 2) annealing at one end of the *PB* transposon was used to generate single-stranded DNA spanning the PB insertion junctions. Then an adapter composed of two oligonucleotides (adaptor-A and adaptor-B, Supplementary Table 2) was ligated to the 3′ end of single-stranded DNA. To enrich DNA spanning the *PB* junctions and add sequences needed for Illumina sequencing, two sequential PCR reactions were conducted using primers that annealed to the end of *PB* and the adaptor, respectively. In the first PCR reaction using an indexed primer (Junction-seq-indexed-primer, Supplementary Table 2) and oligo-498 (Supplementary Table 2), 4-nt indexes were incorporated to allow parallel analysis. In the second PCR reaction, primers seq-f and seq-r (Supplementary Table 2) were used. Single-read sequencing was performed using Illumina HiSeq 2000 or 2500. Raw reads were first filtered by the index and *PB* sequences and then trimmed to keep only the *PB*-flanking genomic DNA sequence. Mapping was performed using Bowtie and only unique and perfect alignments were kept. If the distance of two insertions were less than 3 bp, the insertion with the lower read number was discarded. Identical insertions from different samples of the same query gene were considered as independent events. The reference genome is divided into ORFs and intergenic regions, and G-test was applied to assess the enrichment of *PB* insertions in each region by comparing the observed insertions with potential insertion sites (TTAA sites). The suppressor candidates were chosen based on the P values calculated using the G-test and manual inspection.

**Re-creation and verification T-BOE and C-BOE suppressors.** To re-create the candidate suppressors identified from T-BOE screens, we replaced the ORF of the candidate suppressor gene with a marker gene, or inserted a marker gene at the site of *PB* insertion, or did both.

To re-create the candidate missense mutations identified from C-BOE screens, we either introduced the mutant alleles at the original loci, or in one case (bypass of *cog8* by a *sly1* mutation) integrated a mutant version of the suppressor gene at the *pha2* locus using the pPHA2H vector in a background where the suppressor gene was deleted.

T-BOE suppressors were verified using one of the following two strategies, and C-BOE suppressors were verified using the second strategy.

In the first strategy, the heterozygous essential gene null allele and the heterozygous candidate suppressor allele were introduced into an $h+/h-$ diploid strain. Spores derived from the diploid strain were first germinated without any selection. There existed spores of four different genotypes: WT, the suppressor gene single mutant, the essential gene single mutant, and the double mutant. If the candidate suppressor mutation could rescue the lethality of the essential gene deletion, all spores, except for the essential gene single mutant spores, would form viable colonies. Otherwise, the double mutant spores cannot form colonies either. To discriminate the two cases, the master plates were replica-plated to plates selecting for the marker gene representing the essential gene deletion only, plates selecting for the marker gene representing the suppressor mutant allele only, and plates selecting for the two marker genes both. For a true suppression interaction, we expected that the all the colonies with essential gene null allele should also have the suppressor allele, which can be inferred by comparing the plates.

In the second strategy, a query strain with the essential gene deletion and the rescuing plasmid was crossed to a strain with the candidate suppressor mutant allele. The spores were first germinated without any selection and then the colonies were transferred to plates selecting essential gene deletion and counter-selecting against the rescuing plasmid. For true suppressor mutations, we expected viable colonies on the selection plates. As a negative control, the query strain was also crossed to a wild-type strain.

The verification data can be accessed at https://bypass-of-essentiality.github.io/.

**OP-BOE screens using the Yoshida ORFeome YFH library**. Query strains used for OP-BOE screens were constructed by replacing the *kanMX* marker in T-BOE query strains with the minimal-medium-compatible *natMX* marker. If the mating type of query strains were $h+$, $h-$ strains were obtained by crossing with an $h-$ strain DY11115.

The ORFeome YFH plasmid library, which expresses C terminally YFP-Flag-His$_6$ (YFH)-tagged *S. pombe* ORFs from the strong *Pnmt1* promoter[59], was transformed as a pool into the strain DY6064 ($h+$ his3-D1 leu1-32 ade6-M210 ura4-D18 arg6::PB[ura4+] rpl42::cyhR (SP56Q)) and integrated at the leu1 locus. Leu+ transformants were pooled together. Among the 4910 plasmids presumed to be in the plasmid library, about 94% of them were detected by deep sequencing analysis of the transformant pool.

To perform the screening, equal amounts (20 OD600 units each) of a query strain ($h-$) and the library transformant pool ($h+$) pre-grown in YES to log phase were mixed and washed twice with water. Cells were resuspended in water and spotted onto mating plates, and incubated at 30 °C for three days. The mating mixtures were harvested and digested with glusulase, and spores were purified by centrifugation in Percoll solutions. $1.0 \times 10^7$ spores were germinated in 2 ml of PMG liquid media lacking leucine at 30 °C for 18 h, then $4 \times 10^6$ cells were plated onto 15-cm-diameter PMG−Leu + clonNAT + FUdR plates for selecting progenies without the rescuing plasmid but with the essential gene deletion and an integrated overexpression plasmid. $4 \times 10^5$ cells were plated onto 15-cm PMG−Leu −clonNAT−FUdR plates as input. Plates were incubated at 30 °C for five days, and colonies were harvested for deep sequencing.

**Identifying candidate OP-BOE suppressors by ORF-seq**. The ORFs in the integrated ORFeome plasmids were amplified from genomic DNA by PCR. For sequencing the N terminal sequence of each ORF, PCR primers were ORFseq-up-f and YFP-5-3 (Supplementary Table 2), while for sequencing the C terminal sequence of each ORF, primers were ORFseq-dn-r and LD214 (Supplementary Table 2). N terminal indexed and C terminal indexed PCR products were pooled separately, then a primer extension reaction was performed using the primer seq-f (Supplementary Table 2). ssDNA resulted from the primer extension reaction was ligated to an adaptor composed of two oligos adaptor-A and adaptor-B (Supplementary Table 2). After ligation, a PCR reaction was performed using the primer seq-f and oligo-498 (Supplementary Table 2). PCR products were mixed and DNA in the size range of 250 bp to 500 bp was used for single-read Illumina sequencing on HiSeq 2000 or 2500.

Sequencing reads were assigned to different samples according to their four-nucleotide indexes and assigned to different ORFeome plasmids according to the plasmid-specific-sequence in primers used in constructing the ORFeome library[59]. Reads from 18 independent input samples were combined to use as a control in data analysis.

For most of the query essential genes, three independent OP-BOE screens were performed. Using the Illumina sequencing data of the three screens, we calculated a BOE-score for each ORFeome plasmid. The BOE-score was calculated as:

BOE-score = C-score * $\mu$ (C-score = 0, if $|\mu / \sigma| < 1$; C-score = $1 - 1/|\mu / \sigma|$, if $|\mu / \sigma| \geq 1$). $\mu$ is the average of three weighted average ratios (WA_ratios) of an ORFeome plasmid and $\sigma$ is the standard deviation of the three WA_ratios.

The WA_ratio was calculated as:

WA_ratio = (N_ratio * N_reads + C_ratio * C_reads) / (N_reads + C_reads); where

N_ratio = $\log_2$ ((N_control_reads + 1) / (N_reads + 1) * total_N_reads / total_N_control_reads)

and

C_ratio = $\log_2$ ((C_control_reads + 1) / (C_reads + 1) * total_C_reads / total_C_control_reads).

**Verification of OP-BOE suppressors**. Candidate OP-BOE suppressors were selected according to BOE-scores. To verify the candidates, each candidate YFH-tagged ORFeome plasmid was transformed into a WT strain DY11110 ($h+$ leu1-32 ura4-D18 ade6-M210 arg6::PB[ura4+]), while a control plasmid pDUAL-Yc was transformed into the same strain as a negative control. These transformants were crossed to corresponding query strains and spore suspensions were spotted onto PMG−Leu−thiamine and PMG−Leu + thiamine plates for germination. After incubation at 30 °C for three days, cells were replica-plated to PMG−Leu−thiamine + clonNAT + FUdR and PMG−Leu + thiamine + clonNAT + FUdR plates, respectively. True OP-BOE suppressors were those that allowed cross progenies to grow on PMG−Leu−thiamine + clonNAT + FUdR plates but not on PMG−Leu + thiamine + clonNAT + FUdR plates. Background growth was assessed using the pDUAL-Yc control cross. Using the same procedure, some of the OP-BOE suppressors were verified by random spore analysis where spores were allowed to form individual colonies. In additional to the ORFeome library plasmids, we also performed verification using independently constructed overexpression plasmids expressing tagless ORFs. In one case (gpi3), the YFH-tagged ORFeome plasmid could not be recovered from the library, and thus verification was only performed using the tagless ORF plasmid.

The verification data can be accessed at https://bypass-of-essentiality.github.io/.

**GO slim analysis**. PomBase fission yeast biological process GO slim annotations were downloaded on 13 May 2017 (http://www.geneontology.org/ontology/subsets/goslim_pombe.obo)[62,63]. Term enrichment in the essential, screened, and bypassable gene lists were evaluated using Fisher's exact test.

**Analyzing gene features**. (1) Analysis of paralogs. The paralog relationship was obtained by intersecting the paralog list from Kim et al.[55] and the paralog list from Ensembl BioMarts. (2) The dispensability of *S. cerevisiae* orthologs. The ortholog relationship between genes in *S. pombe* and *S. cerevisiae* was based on PomBase ortholog curation downloaded on 11 April 2017 (ftp://ftp.pombase.org/pombe/orthologs/cerevisiae-orthologs.txt)[62]. The dispensability information of *S. cerevisiae* genes was inferred from lists of genes with viable phenotype and genes with inviable phenotype downloaded from SGD on 27 April 2017 (https://www.yeastgenome.org/phenotype/viable and https://www.yeastgenome.org/phenotype/inviable). In the two lists, only S288C entries from systematic analyses using null alleles were used. (3) Slow spore lethality. Phenotype of germinating deletion mutant haploid spores derived from heterozygous deletion diploids was based on a systematic visual analysis of a genome-wide fission yeast gene deletion library[23]. We categorized the deletion phenotype into two types according to column G of Supplementary Table 1 in Hayles et al.[23]: rapid lethality for spores that failed to germinate and spores that germinated but did not form microcolonies, slow lethality for spores that germinated and formed microcolonies or small colonies. (4) *Hermes* insertional profile. The transposon *Hermes* insertional profiles were from ref. [24]. (5) Gene expression level. The mRNA and protein abundances of *S. pombe* genes were from ref. [64]. (6) Protein-degradation rate. The degradation rates of *S. pombe* proteins were from ref. [65]. (7) Gene evolutionary rate. The gene evolutionary rates calculated based on the branch lengths of gene trees constructed using orthologous genes of four *Schizosaccharomyces* species were from ref. [41]. (8) Codon adaptation index. Codon adaptation index (CAI) values were downloaded from PomBase on 12 April 2017 (ftp://ftp.pombase.org/pombe/Protein_data/PeptideStats.tsv)[62]. (9) tRNA adaptation index. tRNA adaptation index (tAI) values were from ref. [66]. (10) Number of Ascomycota species harboring an ortholog. Based on the orthology information for 23 Ascomycota species at the Fungal Orthogroups Repository (https://portals.broadinstitute.org/regev/orthogroups/), the numbers of species with an ortholog of an *S. pombe* gene were used as a measure for ortholog conservation in Ascomycota. (11) Broad conservation index. The number of genomes encoding genes belonging to the same InParanoid-defined ortholog group as an *S. pombe* gene was used as a measure for the broad phylogenetic conservation of that gene. (12) Other features of *S. pombe* genes. The ratio between nonsynonymous and synonymous changes (dN/dS), the disorder index representing the percentage of unstructured residues, the effective number of codons used (Nc), the co-expression degree, the expression variation, and the multifunctionality score representing the number of associated GO annotations were from ref. [67].

**Analyzing features of essential protein complexes**. (1) List of protein complexes containing subunits encoded by essential genes. The list of protein complexes containing subunits encoded by essential genes (Supplementary Data 3) was mainly based on the list of macromolecular complexes downloaded from PomBase on 22 March 2017 (ftp://ftp.pombase.org/pombe/annotations/Gene_ontology/GO_complexes/Complex_annotation.tsv)[62,63]. Out of 253 complexes that each contains at least one subunit encoded by an essential gene, we manually recompiled a list of 146 complexes that cover 97% (689/710) of essential genes in the original 253 complexes. To expand the coverage of essential genes, we added 37 physically interacting protein pairs and trios (denoted as 'binding_module' in Supplementary Data 3). They are based on "protein binding" GO annotation (GO:0005515) and only reciprocally annotated gene pairs and trios were included[62,63]. (2) Assessing whether bypassability is shared by complex subunits. 20 protein complexes with at

least two subunits encoded by chrII-L essential genes were categorized into three types: 5 bypassable complexes, 11 non-bypassable complexes, and 4 mixed complexes (Supplementary Data 3, column G and H). Permutation analysis was used to assess which types are over- or under-represented. We randomly assigned the same number of bypassable essential genes to the 20 protein complexes. Using the randomization result, we re-defined the bypassability of the 20 protein complexes. By repeating this process 1000 times, we estimated the expected number of each type of protein complexes and obtained $P$ values. (3) Identifying protein complex features that can predict bypassability. A total of 19 quantitative features of protein complexes were analyzed. Four features were analyzed using percentage values: percentage of essential genes, percentage of essential genes with non-essential *S. cerevisiae* ortholog or without *S. cerevisiae* ortholog, percentage of essential genes harboring orthologs in 22 other Ascomycota genomes, and percentage of essential genes with slow spore lethality. The other 15 features, analyzed as medians, include: the evolutionary rate, the broad conservation index, the number of other Ascomycota genomes harboring orthologs, dN/dS, the mRNA abundance, *Hermes* insertional density, the disorder score, the protein abundance, the effective number of codons used (Nc), the co-expression degree, the protein-degradation rate, the mRNA half-life, the protein length, the expression variation, and the multi-functional score of essential subunits. We used the Wilcoxon signed-rank test to determine which complex features can distinguish bypassable complexes from non-bypassable complexes. In addition, we calculated the area under the receiver operating characteristic curve (AUC) using pROC[68] to evaluate the predictive power of the complex features. Only bypassable complexes and non-bypassable complexes were included in the AUC analysis. (4) Hierarchical clustering of essential protein complexes. A total of 127 protein complexes with at least two subunits encoded by essential genes were analyzed by hierarchical clustering. Euclidean distance and average linkage clustering method were used. Complex features with significant power to predict bypassability were analyzed.

**Datasets used to assess gene functional similarity**. (1) Datasets used to assess whether genes acting in same biological processes. Genes sharing at least one GO slim annotation were considered to act in same biological processes, and genes that each has at least one GO slim annotation but do not share any GO slim annotation were considered to act in different processes. (2) Datasets used to assess whether proteins co-localize. Protein localization data were obtained from GO association entries derived from ref. [59]. Proteins sharing at least one subcellular localization annotation were classified as co-localizing. Proteins only sharing annotations of nucleus localization (GO:0005634) or cytosol localization (GO:0005829) were not considered co-localizing due to the low specificity of these two terms. Two proteins that each has annotated localization(s) but do not share any were classified as not co-localizing. (3) Datasets used to assess whether proteins physically interact. Protein-protein physical interactions were extracted from the BioGRID database (3.4.147 version)[69].

**Analyses on the enrichment of functionally similar genes**. Negative and positive genetic interactions were from a genome-scale epistasis map (E-MAP) analysis[41] based on S score cutoffs used by the authors of that study (S score < −2.3 and S score > 1.8 for negative and positive interactions, respectively). The E-MAP study was performed by crossing 862 query strains that are mostly deletions to 1955 array strains that are all deletions. To make a fair comparison, we only considered BOE interactions with null alleles of non-essential genes as suppressors.

To analyze the enrichment of functionally similar genes among interacting gene pairs, we compared the interacting gene pairs against a background set of gene pairs. For the E-MAP data, the background was all assayed gene pairs. For the BOE data, the background was all pair-wise combinations between one of the 142 screened essential genes and one of 3578 non-essential genes.

To analyze the enrichment of functionally similar genes among interactor-sharing gene pairs, we compared the interactor-sharing pairs (query-sharing suppressor gene pairs for the BOE data) against a different background set of gene pairs. For the E-MAP data, the background was all pair-wise combinations among 1955 array genes. For the BOE data, the background was all pair-wise combinations among the 3578 non-essential genes.

**Induction of mitochondrial DNA loss using ethidium bromide**. Cells were first grown on PMG solid medium lacking thiamine to allow the expression of OP-BOE suppressor genes, and then streaked onto PMG medium without thiamine but supplemented with 12.5 μg/ml of ethidium bromide and 2% of potassium acetate[28]. After growth at 30 °C without light for six days, cells were streaked onto PMG medium without thiamine. Nine days later, small colonies were restreaked onto PMG medium without thiamine. Five days later the loss of mitochondrial DNA was verified by PCR (primers are listed in Supplementary Table 2).

**Strand-specific RNA sequencing**. mRNA was isolated from total RNA using the Dynabeads mRNA Purification Kit, fragmented by 5x first strand buffer, and reverse-transcribed using the SuperScript II Reverse Transcriptase. First-strand DNA/RNA hybrid was precipitated with ammonia acetate and glycogen to remove free dNTPs. Second-strand cDNA was synthesized in the second-strand buffer using dNTPs with dTTP replaced by dUTP, RNase H, DTT, and DNA Polymerase

I, and purified using the Agencourt AMPure XP beads. Sequencing libraries were prepared using NEBNext DNA Library Prep Master Mix Set for Illumina following its instruction with an additional step of USER enzyme treatment after adaptor ligation. These libraries were then analyzed with Illumina HiSeq 2500 in the single-read mode. Sequencing data have been deposited at NCBI SRA under accession SRP142375.

Sequencing reads were mapped to the reference genome based on the annotation file Schizosaccharomyces_pombe.ASM294v1.18.gtf using TopHat2 with options --library-type fr-firststrand -i 28 -I 820 --min-segment-intron 28 --min-coverage-intron 28. Differential expression analysis was performed using DESeq.

***lacZ* reporter assay to measure DSR-mediated degradation**. To create *lacZ* reporter plasmids, the fragment of *act1* promoter followed by a *lacZ* ORF from pART-*lacZ* was cloned into the pPHA2H vector to get pPHA2H-*lacZ* (pDB2064). A region in *mei4* ORF (486-828 bp), which contains several DSR motifs, was inserted after the AscI site immediately following the stop codon of *lacZ* ORF to get pPHA2H-*lacZ*-DSR (pDB3844). The reporter plasmid was integrated at the *pha2* locus and the expression level of the reporter gene was determined using the β-galactosidase activity assay.

**Yeast two-hybrid analysis**. For yeast two-hybrid analysis, we used the Matchmaker system (Clontech). cDNAs of prey genes and bait genes were cloned into a modified pGAD GH vector and a modified pGADT7 vectors (Clontech), respectively. Bait and prey plasmids were co-transformed into the AH109 strain and transformants were selected on the double dropout medium (SD/-Leu/-Trp). The activation of the *HIS3* and *ADE2* reporter genes was assessed on the quadruple dropout medium (SD/-Ade/-His/-Leu/-Trp).

**Immunoprecipitation**. About 100 OD600 units of log-phase cells were collected and lysed by glass bead beating in the lysis buffer (50 mM Tris, pH 7.5, 1 mM EDTA, 150 mM NaCl, 10% glycerol, 1 mM PMSF, 1 mM DTT, 0.05% NP-40, 1 × protease inhibitor cocktail). After the cell lysate was cleared by centrifugation, the supernatant was inoculated with the GFP-trap agarose beads. After washing the beads with the lysis buffer three times, proteins bound to beads were eluted with the SDS-PAGE loading buffer.

**Histone peptide pull-down assay**. N-terminally HIS$_6$-GST-tagged WT and mutated Png1 proteins were purified from the *Escherichia coli* BL21 strain. Biotinylated histone peptides were purchased from Merck Millipore. 1 μg of Png1 proteins were incubated with 1 μg of histone peptides in 300 μl of binding buffer (50 mM Tris-HCl, pH 7.5, 300 mM NaCl, 0.1% NP-40, 1 mM PMSF) at 4 °C overnight. The Streptavidin Sepharose beads were used to pull down the histone peptides. After washing with the binding buffer three times, proteins bound to beads were eluted by boiling in the SDS-PAGE loading buffer.

**Global SUMOylation analysis**. *slx8-29* strains and other strains were grown in YES media at 25 °C for 24 h, then they were diluted and transferred to 37 °C to grow for 12 more hours. 1.2 OD600 units of cells were collected, washed once with ddH$_2$O, and lysed in 100 μl of 1.85 M NaOH and 7.4% β-mercaptoethanol on ice for 5 min. Then 100 μl of 50% TCA was added and precipitates were pelleted at 15000 rpm at 4 °C for 2 min. The pellet was washed once with 400 μl of ice-cold acetone and pelleted at 15000 rpm at 4 °C for 5 min, air dried, and was then resuspended with 60 μL of HU buffer (8 M urea, 200 mM Tris-HCl, pH 6.8, 1 mM EDTA, 5% (w/v) SDS, 0.1% (w/v) bromophenol blue, 1.5% (w/v) dithiothreitol). Samples were denatured at 60 °C for 10 min before loading onto a three-layer NuPAGE gel composed of 4% stacking gel, and a separating gel consist of 6% separating gel followed with 12% separating gel. SUMO conjugates were detected by immunoblotting using an anti-Pmt3 rabbit polyclonal antibody. Uncropped and unprocessed scans of the blots are shown in the Source Data file.

**Affinity purification coupled with mass spectrometry**. Two thousand OD600 units of cells cultured in PMG liquid media were collected. GFP-trap-based purification and mass spectrometry analysis were performed[70]. Cells were lysed in equal volume lysis buffer (50 mM HEPES, pH 7.5, 150 mM NaCl, 1 mM EDTA, 1 mM DTT, 1 mM PMSF, 0.05% NP-40, 10% glycerol, 1 × Roche protease inhibitor cocktail) by bead beating. Lysates were centrifuged and supernatants were incubated with GFP-trap agarose beads at 4 °C. After washing with lysis buffer without NP-40, beads-bound proteins were eluted with elution buffer (1% SDS, 100 mM Tris, pH 8.0) at 80 °C and were precipitated with 20% TCA. Precipitates were washed with acetone and then dissolved in 8 M urea, 100 mM Tris-Cl, pH 8.5. After reducing with 5 mM TCEP and alkylating with 10 mM iodoacetamide, samples were digested with trypsin. For LC-MS/MS analysis, the digested peptides were loaded to a precolumn (100 μm ID, 4 cm in length, packed with C12 10 μm 120 Å resin from YMC) and separated on an analytical column (75 μM ID, 10 cm in length, packed with Luna C18 3 μm 100 Å resin from Phenomenex) by an Easy-nLC II HPLC (Thermo Fisher Scientific) system coupled to an LTQ Orbitrap XL mass spectrometer (Thermo Fisher Scientific). The MS2 spectra were searched with

Prolucid against an S. pombe protein database. The search results were filtered by DTASelect.

**Microscopy**. Microscopy was performed using a DeltaVision PersonalDV system (Applied Precision) equipped with a CFP/YFP/mCherry filter set (Chroma 89006 set). Quantitative analysis of the GFP foci was performed using the Pixel_Inspection_Tool.java of Image J.

**Query strains for double mutant suppressor screens**. The rescuing plasmid pPC96-LEU2-Prhb1-Rqh1, in which the expression of rqh1 is driven by the rhb1 promoter, was introduced into ZXRY10 (h- leu1-32 ura4-D18 ade6-M210 arg6::PB [ura4+] rhq1Δ::natMX) by transformation. Transformants were selected on PMG plates without leucine and adenine. The episomal state of the plasmid was confirmed as described above. The confirmed strain was crossed with strains harboring a kanMX-marked deletion mutation synthetic-lethal with rqh1Δ (h + leu1-32 ura4-D18 ade6-M210 arg6::PB[ura4+] xxxxΔ::kanMX). Spores were allowed to germinate on YES plates. Colonies formed on YES plates were replica plated to low-adenine YE + G418 + clonNAT plates. Pink colonies were chosen for the further confirmation of the episomal state of the plasmid as described above, and confirmed clones were used as query strains for T-BOE screens.

**Reporting summary**. Further information on experimental design is available in the Nature Research Reporting Summary linked to this article.

## Data availability

RNA-seq data have been deposited at NCBI SRA under accession SRP142375. The experimental data that verify BOE interactions listed in Supplementary Data 2 can be accessed at https://bypass-of-essentiality.github.io/. The source data underlying Supplementary Figs. 7b, 7g, 7j, 7k, and 7m are provided as a Source Data file.

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

## Acknowledgements

We thank Katsunori Tanaka for an unpublished *GFP-pmt3* strain, Thomas D. Fox for the *ptp1-1* strains, Minoru Yoshida and RIKEN DNA Bank for the plasmid vectors and ORFeome clones, NIBS Sequencing Center for mRNA sequencing, Jim Haber, Josh Mitteldorf, Wenfeng Qian, and Paul Russell for suggestions on improving the manuscript, and John Hugh Snyder for editing the manuscript and helping sharpen its argument and structure. This work was supported by National Natural Science Foundation of China (Grant No. 31571290) and supports from the Ministry of Science and Technology of China and the Beijing municipal government.

## Author contributions

Conceptualization: J.L., H.-T.W., W.-T.W., and L.-L.D.; Methodology and investigation: J.L., H.-T.W., W.-T.W., X.-R.Z., J.-Y.R., Y.B., Y.-X.X., W.H., and L.-L.D.; Formal analysis: J.L., H.-T.W., F.S., M.-Q.D., and L.-L.D.; Writing – original draft: J.L., H.-T.W., and L.-L.D.; Writing – review and editing: J.L., H.-T.W., M.-Q.D., and L.-L.D.; Funding acquisition: M.-Q.D. and L.-L.D.

## Additional information

**Competing interests:** The authors declare no competing interests.

