## [Peer Review File · Nature Communications]

Reviewers' Comments:

Reviewer #1:

Remarks to the Author:

In this paper, the authors applied a systematic methodology (termed as BOE analysis) to identify genetic factors that could suppress the lethal effect of knocking down certain essential genes. The essential genes for which they could identify such suppressors, the authors classify them as bypassable essential genes. By comparing the evolutionary rate, codon optimality and the phylogenetic distribution between bypassable and non-bypassable essential genes, the authors conclude that the bypassability is associated with low gene importance. By analyzing the essentiality of the by-passable essential genes in other species of yeast they draw a strong relationship between the bypassability and differential essentiality. By analyzing the role of identified suppressors in bypassing the gene essentiality, the authors propose three possible essentiality-bypassing mechanisms. By detailing some case studies the authors also demonstrate that the BOE analysis can be used to infer gene functions. Overall, this is an interesting study and the manuscript is well written. However, the following comments should be addressed to strengthen this manuscript.

Comments

Major

- Page 5, line 6: Can the authors provide more details on why 11% of the essential genes on chrII-L were dropped out of the bypass of essentiality study? Ideally, the method should be generalized enough to assess all the genes.
- Page 8, line 27: The authors generalize that the constituent subunits of given essential protein complex tend to be either all bypassable or all non-bypassable. However, the authors do find complexes that are made up of both bypassable and non-passable subunits (mixed complexes). Though the occurrence of such mixed complexes is less in the set of the complexes considered for this study, their occurrence might be more in the largest set in the biological system as a whole. Thus the mixed complexes might not be "inconsistent cases" as the authors claim in line 20-22 of page 9 and the bypassability might not be a protein complex level property.
- Page 11, line 16-19: The authors should explain in detail in the text at least one example where the suppressor acted in a paralog dependent manner.
- Page 11, line 20-28: The classification of essentiality-bypassing mechanisms is not clearly analyzed even though this could be the most interesting section of the paper. The authors should provide the analysis of all the bypassable genes in Fig. 2 instead of only a few examples in supplementary Fig. 6. More importantly, many studies have shown that cells will evolve with seemingly unrelated functions to rescue the deletion of essential genes [1,2]. However, the classification by the authors seems to focus on the types of strategies that cells use to fix the broken machinery. In other words, are there any relationships in Fig. 2 that does not fall into this classifications?
- Page 12, line 14-16 and Fig. 6. The fold enrichment over expected value (>150) In Fig. 6G is way larger than the value (>30) in Fig. 6D. Does this imply that the suppressors interact more frequently with each other than with the query essential gene? Can the authors comment on the large differences between the values?

Minor

- Page 9, line 29: To which data do the authors refer to by mentioning "Our data suggest that..."? Please provide the reference (to figures or tables)
- Page 14, line 8: it is better to change the section title from "non-essential genes" to "conditional-essential genes" because all the non-essential genes that can be evaluated by this method have to first become essential in a mutant strain.
- Page 5: line 23-26: Rancati et al. have reviewed and proposed a quantitative definition of gene essentiality [3]. The authors may consider comparing their definition of gene importance with their

definition of “low essentiality” and “high essentiality”.

References

1. Liu G, Yong MYJ, Yurieva M, Srinivasan KG, Liu J, Lim JSY, et al. Gene Essentiality Is a Quantitative Property Linked to Cellular Evolvability. *Cell*. 2015;163: 1388–1399.
2. Patrick WM, Quandt EM, Swartzlander DB, Matsumura I. Multicopy suppression underpins metabolic evolvability. *Mol Biol Evol*. 2007;24: 2716–2722.
3. Rancati G, Moffat J, Typas A, Pavelka N. Emerging and evolving concepts in gene essentiality. *Nat Rev Genet*. 2018;19: 34–49.

Reviewer #2:

Remarks to the Author:

The authors exploited advantages of yeast genetics to investigate meaning of gene essentiality. Through this manuscript, by focusing on 148 genes on the chromosome II left arm annotated as essential, they are able to categorise into two types of “essential” genes, possible to bypass or not. Logic of genetic complementation is well summarised. Whereas these suppressor screenings are not novel, they were able to predict gene bypassability by studying the phenotype of the lethality. Their studies established theories and the system for bypassing essential genes and would be useful widely for the pombe community. Their analysis is also useful for evolutionary biologists. Their BOE analysis also elucidated a group of genes working as a complex or pathways. Pilot studies (such as *rad21Δ* and *slx8Δ* suppressors,) are also well documented and are useful for specific area of scientists. Collectively, this is well written and documented paper. I recommend publication of this manuscript. I have only minor questions below.

1. How many cells are screened per assay (for C-BOE, T-BOE and OP-BOE)? How was statistical power calculated to endure the screening covered adequately? Obviously, this strategy is useful for the screening of gene group that complement a mutant of interest. would help readers to replicate the study.
2. Where are generated constructs (yeast strains, plasmids, libraries for the screening) available or deposited?
3. Figure 2 is not well explained in main text (page 5 line 11).

Reviewer #1

Comments

Major

● Page 5, line 6: Can the authors provide more details on why 11% of the essential genes on chrII-L were dropped out of the bypass of essentiality study? Ideally, the method should be generalized enough to assess all the genes.

Response: Out of the 159 essential genes on chrII-L, no BOE analysis results were obtained for 17 (11%) genes. One of these 17 essential genes, the telomere protection gene *stn1*, could not be analyzed because its essentiality can be bypassed at a high frequency by chromosome circularization (Martin et al. 2007). For the other 16 genes (*gnd1*, *mrps16*, *usp107*, *ubc4*, *orm1*, *pre4*, *git7*, *gta1*, *med15*, *cct8*, *gtr1*, *sec66*, *trz2*, *kap109*, *toa1*, and *rpb1*), we failed to obtain their query strains using our current query strain construction strategy, which employs an episomal “rescuing plasmid” to express the query gene. The reasons behind the strain construction failure are not clear. We speculate that one possible reason may be that cells cannot tolerate abnormal expression levels of these query genes. Supporting this idea, three of these 16 genes, *ubc4*, *trz2*, and *rpb1*, have been shown in the literature to cause cell inviability when overexpressed (Arita et al. 2011; Gan et al. 2011; Javerzat et al. 1996). Such a problem can in theory be tackled through changing the promoter driving the query gene expression in the rescuing plasmid. In addition, we are currently developing a plasmid-free query strain construction strategy that can keep the query gene expression unaltered. In the revised manuscript, we added two sentences in the main text to provide more details on these unanalyzed genes.

Martin, V., Du, L. L., Rozenzhak, S. & Russell, P. Protection of telomeres by a conserved Stn1-Ten1 complex. Proc Natl Acad Sci U S A 104, 14038-14043, doi:10.1073/pnas.0705497104 (2007).

Arita, Y. et al. Microarray-based target identification using drug hypersensitive fission yeast expressing ORFeome. *Mol Biosyst* 7, 1463-1472, doi:10.1039/c0mb00326c (2011).

Gan, X. et al. The fission yeast *Schizosaccharomyces pombe* has two distinct tRNase Z(L)s encoded by two different genes and differentially targeted to the nucleus and mitochondria. *Biochem J* 435, 103-111, doi:10.1042/BJ20101619 (2011).

Javerzat, J. P., Cranston, G. & Allshire, R. C. Fission yeast genes which disrupt mitotic chromosome segregation when overexpressed. *Nucleic Acids Res* 24, 4676-4683 (1996).

● Page 8, line 27: The authors generalize that the constituent subunits of given essential protein complex tend to be either all bypassable or all non-bypassable. However, the authors do find complexes that are made up of both bypassable and non-passable subunits (mixed complexes). Though the occurrence of such mixed complexes is less in the set of the complexes considered for this study, their occurrence might be more in the largest set in the biological system as a whole. Thus the mixed complexes might not be “inconsistent cases” as the authors claim in line 20-22 of page 9 and the bypassability might not be a protein complex level property.

Response: We agree with the reviewer that incomplete sampling prevents us from definitively ascertaining the extent of the essential subunits of the same protein complex being either all non-bypassable or all bypassable. Thus, we have removed our claim that “bypassability is a protein-complex-level property”, and instead, draw a much weaker conclusion that “subunits belonging to the same protein complex tend to share bypassability”. We believed that such a conclusion is justified by our permutation analysis based on the chrII-L BOE data (Fig. 4a) and our follow-up analyses of three complexes (Supplementary Figs. 3c-e). We have also revised the “inconsistent cases” sentence, changing it to “In cases where subunits of the same protein complex do not

share bypassability or BOE suppressors, new insights on the functional differences between subunits can be gained.”

- Page 11, line 16-19: The authors should explain in detail in the text at least one example where the suppressor acted in a paralog dependent manner.

Response: We thank the reviewer for pointing out that we should explain in detail at least one example of paralog-dependent suppressors. For the two cases shown in Fig. 5b, we do not know the exact mechanistic relationships between the suppressor gene and the paralog gene, and thus cannot provide truly satisfactory explanations. However, in a later section under the subheading “**Inferring gene functions with BOE analysis**”, we present a case in which a non-paralog suppressor (*erh1Δ*) bypasses an essential gene (*rad21*) in a paralog-dependent way, and we can clearly explain the relationship between the suppressor gene (*erh1*) and the paralog gene (*rec8*). We have now added the following sentences to offer hypothetical and broad-stroke explanations on the two cases shown in Fig. 5b and alert the readers that a case with a more concrete and detailed explanation will be presented later: “To what extent this model can explain the suppressor-paralog relationships in the two cases shown in Fig. 5b awaits further analysis. It is possible that some paralog-dependent suppressors may not act through activating the paralog, but rather through reducing the need for the function of the essential gene, thus allowing a weak backup activity to become sufficient for supporting viability. An example that mechanistically conforms to the model of activating a dormant redundant paralog will be given below.” In the section where the paralog-dependent bypass of *rad21* by *erh1Δ* is presented, we also added data showing a triple mutant analysis of *rad21Δ*, *erh1Δ*, and *rec8Δ* as Supplementary Fig. 7h in the revised manuscript.

● Page 11, line 20-28: The classification of essentiality-bypassing mechanisms is not clearly analyzed even though this could be the most interesting section of the paper. The authors should provide the analysis of all the bypassable genes in Fig. 2 instead of only a few examples in supplementary Fig. 6. More importantly, many studies have shown that cells will evolve with seemingly unrelated functions to rescue the deletion of essential genes [1,2]. However, the classification by the authors seems to focus on the types of strategies that cells use to fix the broken machinery. In other words, are there any relationships in Fig. 2 that does not fall into this classifications?

Response: We thank the reviewer for raising this important point. We agree with the reviewer that a truly comprehensive understanding of bypassing mechanisms demands mechanistic explanations of most if not all of the BOE interactions. Unfortunately, we can satisfactorily explain at the molecular level only a small minority of the BOE interactions found in our study, and these explainable cases are all presented in Supplementary Fig. 6. We have now added in the main text the following sentences and cited the two studies mentioned by the reviewer: “We note that most of the BOE interactions uncovered in this study cannot yet be classified into one of these mechanism types, owing to a shortage of molecular-level evidence on the query-suppressor relationships. It is possible that a substantial fraction of BOE interactions occur through other types of mechanisms, including mechanisms seemingly unrelated to the functions of the query genes (Liu et al. Cell 2015 and Patrick et al. MBE 2007).”

● Page 12, line 14-16 and Fig. 6. The fold enrichment over expected value (>150) in Fig. 6G is way larger than the value (>30) in Fig. 6D. Does this imply that the suppressors interact more frequently with each other than with the query essential gene? Can the authors comment on the large differences between the values?

Response: We agree with the reviewer’s interpretation of the enrichment ratio difference. To emphasize this interesting difference and to provide an explanation, we

have added the following sentences: “Interestingly, interactor-sharing pairs are enriched with physically interacting proteins to a considerably greater extent than BOE interacting pairs, probably because suppressor genes for the same query often include genes encoding multiple subunits of the same protein complex.”

Minor

● Page 9, line 29: To which data do the authors refer to by mentioning “Our data suggest that...” ? Please provide the reference (to figures or tables)

Response: We were referring to the chrII-L BOE analysis data, which show that one Prp19 complex subunit (Cwf15) is bypassable and two other Prp19 complex subunits (Cwf7 and Prp5) are non-bypassable. We have revised the sentence. It now reads: “For the fourth complex, the Prp19 complex, our findings that one of its subunits, Cwf15, is bypassable, whereas two other subunits (Cwf7 and Prp5) are non-bypassable, reveal previously unappreciated functional difference between its subunits.”

● Page 14, line 8: it is better to change the section title from “non-essential genes” to “conditional-essential genes” because all the non-essential genes that can be evaluated by this method have to first become essential in a mutant strain.

Response: We thank the reviewer for the suggestion. We have made the change suggested by the reviewer.

● Page 5: line 23-26: Rancati et al. have reviewed and proposed a quantitative definition of gene essentiality [3]. The authors may consider comparing their definition of gene importance with their definition of “low essentiality” and “high essentiality” .

Response: The four different extents of gene essentiality defined in Rancati *et al.* 2018, which include “no essentiality”, “low essentiality”, “high essentiality”, and “complete essentiality”, are based on either “context dependency” or “evolvability following gene inactivation”. The definitions based on evolvability are more related to “gene bypassability” defined in our manuscript than to “gene importance”. We have now added a new paragraph in the Discussion section to compare the concept of gene bypassability with the quantitative definitions of gene essentiality proposed in Rancati *et al.* 2018.

Reviewer #2

1. How many cells are screened per assay (for C-BOE, T-BOE and OP-BOE)? How was statistical power calculated to ensure the screening covered adequately? Obviously, this strategy is useful for the screening of gene groups that complement a mutant of interest. Would help readers to replicate the study.

Response: For C-BOE screens, we conducted at least two independent screens for each query essential gene, and used the mutagen MNNG to treat approximately 1×10^8 cells in each screen. The MNNG treatment resulted in a viability drop to about 10%. Our genome resequencing of MNNG-mutagenized clones showed that, MNNG induces exclusively nucleotide substitutions (about 90% GC-to-AT transitions and about 10% AT-to-GC transitions), and on average, each mutagenized genome has approximately 40 genes harboring MNNG-induced missense mutations and approximately 1 non-essential gene harboring MNNG-induced nonsense mutations. Because there are about 5000 genes (including around 3600 non-essential genes) in the fission yeast genome, the total pool of 1×10^7 cells that survived the MNNG treatment in each screen should in theory contain, for an average gene, around 80000 independent missense mutations, and for an average non-essential gene, around 2800 independent nonsense mutations. Such numbers of mutations should be large enough to cover most of the coding sequence changes that can be induced by MNNG. An empirical support for the high coverage of C-BOE screens is our observations that special missense mutations were independently recovered more than once in the screens. In the C-BOE screens for *pdf1* suppressors, we isolated the same *sec24-A632T* mutation three times independently, and in the C-BOE screens for *erg27* suppressors, we isolated the same *erg26-R304H* mutation twice independently.

For T-BOE screens, we used about 8×10^5 *piggyBac* transposon-mutagenized cells for each screen and for each query gene, at least two independent screens were conducted. There are approximately 110000 *piggyBac* targeting sequences (TTAA sites) in the fission yeast genome and about 40% of them are in coding sequences. More than

98% of the fission yeast genes contain at least one TTAA site in their coding sequences. Based on our previously published study (Li et al. NAR 2011 PMID:21247877), 8×10^5 *piggyBac*-mutagenized cells should be enough to ensure that most of the TTAA-containing genes have at least one transposon insertion.

For OP-BOE screens, 4×10^6 overexpressing-plasmid-containing cells were used for each screen, and such a cell number should be more than enough to cover most of the around 4900 overexpressing plasmids in the library. Empirically, we found in pilot OP-BOE screens for *slx8* suppressors that using ten times less cells still allowed the identification of all *slx8* OP-BOE suppressors.

We have added more details in the Methods section to ensure that readers can understand and replicate our screening methodology.

2. Where are generated constructs (yeast strains, plasmids, libraries for the screening) available or deposited?

Response: We have added the following sentence in the Methods section: "All plasmids and fission yeast strains generated in this study are available upon request."

3. Figure 2 is not well explained in main text (page 5 line 11).

Response: We have added more explanatory text for Figure 2, as suggested by the reviewer.

Reviewers' Comments:

Reviewer #1:

Remarks to the Author:

Satisfied with the replies and changes.

Reviewer #2:

Remarks to the Author:

The revised manuscript addressed all enquiries adequately. I found that the article is greatly improved in this revision. I fully support publication of this manuscript at Nature Communications.